# A quantitative model of conserved macroscopic dynamics predicts future motor commands

**Connor Brennan[1†], Alexander Proekt[2†]***

[1]Departmentof Neuroscience, University of Pennsylvania, Philadelphia, United States; [2]Department of Anesthesiology and Critical Care, University of Pennsylvania, Philadelphia, United States

**Abstract** In simple organisms such as *Caenorhabditis elegans*, whole brain imaging has been performed. Here, we use such recordings to model the nervous system. Our model uses neuronal activity to predict expected time of future motor commands up to 30 s prior to the event. These motor commands control locomotion. Predictions are valid for individuals not used in model construction. The model predicts dwell time statistics, sequences of motor commands and individual neuron activation. To develop this model, we extracted loops spanned by neuronal activity in phase space using novel methodology. The model uses only two variables: the identity of the loop and the phase along it. Current values of these macroscopic variables predict future neuronal activity. Remarkably, our model based on macroscopic variables succeeds despite consistent inter-individual differences in neuronal activation. Thus, our analytical framework reconciles consistent individual differences in neuronal activation with macroscopic dynamics that operate universally across individuals.

DOI: https://doi.org/10.7554/eLife.46814.001

**\*For correspondence:**
proekt@gmail.com

[†]These authors contributed equally to this work

**Competing interests:** The authors declare that no competing interests exist.

## Introduction

Advances in neuronal imaging (*Kato et al., 2015*; *Ahrens et al., 2013*; *Berényi et al., 2014*; *Jorgenson et al., 2015*; *Venkatachalam et al., 2016*; *Nguyen et al., 2016*; *Schrödel et al., 2013*) are now making it possible to simultaneously record activity in a large number of neurons simultaneously during execution of behaviors. Most analytic techniques used to simplify such complex data-sets involve dimensionality reduction (*Kato et al., 2015*), clustering (*Venkatachalam et al., 2016*), correlations between activity of neuronal populations and behavior (*Georgopoulos et al., 1986*) or features of the sensory stimuli (*Luo et al., 2014*), and connectivity among neurons (*Varshney et al., 2011*). Although having sufficiently detailed experimental observations is absolutely essential, even when analyzed using these sophisticated statistical techniques, detailed information about activation of individual neurons does not always automatically lead to greater understanding of the laws that give rise to the temporal evolution of neuronal activity or the relationship between neuronal activity and the 'computations' performed by the brain (*Frégnac, 2017*; *Jonas and Kording, 2017*).

Most modeling approaches aimed at understanding how the observed neuronal activity unfolds in time proceed in a bottom-up fashion. In simple nervous systems, such as the stomatogastric nervous system (*Hartline, 1979*), feeding central pattern generator in *Aplysia* (*Susswein et al., 2002*), and locomotor circuitry in nematode *Caenorhabditis elegans* (*Kunert et al., 2014*) realistic models built on biophysics of individual neurons and properties of their connections can be constructed. Attempts have been made to model more complex neural networks such as a cortical column at the level of biophysical properties of individual neurons (*Markram, 2006*; *Markram et al., 2015*). Although these modeling approaches can prove successful in some settings, the bottom-up

**eLife digest** How can we go about trying to understand an object as complex as the brain? The traditional approach is to begin by studying its component parts, cells called neurons. Once we understand how individual neurons work, we can use computers to simulate the activity of networks of neurons. The result is a computer model of the brain. By comparing this model to data from real brains, we can try to make the model as similar to a real brain as possible.

But whose brain should we try to reproduce? The roundworm *C. elegans*, for example, has just 302 neurons in total. Advances in brain imaging mean it is now possible to identify each of these neurons and compare its activity across worms. But doing so reveals that the activity of any given neuron varies greatly between individuals. This is true even among genetically identical worms performing the same behavior.

Researchers trying to model the roundworm brain have attempted to model the average activity of each neuron across many worms. They hoped they could use these averages to predict the behavior of other worms from their neuronal activity. But this approach did not to work. Even in roundworms, the coordinated activity of many neurons is required to generate even simple behaviors. Averaging the activity of neurons across worms thus scrambles the information that encodes each behavior.

Brennan and Proekt have now overcome this problem by developing a more abstract model that treats the nervous system as a whole. The model takes into account changes in the activity of neurons, and in the worms' behavior, over time. A model of this type built using one set of worms can predict the behavior of another set of worms. This approach may work because in evolution natural selection acts at the level of behaviors, and not at the level of individual neurons. The activity of individual neurons can thus vary between animals, even when those neurons encode the same behavior. This means it may also be possible to model the human brain without knowing the activity of each of its billions of neurons.

DOI: https://doi.org/10.7554/eLife.46814.002

approach is limited in several fundamental ways. Even in the simplest nervous systems biophysically realistic models can rarely be sufficiently constrained by the available experimental measurements (*Selverston, 1980*). Biophysical properties of individual neurons and their connections change dynamically as a function of neuromodulation and neuronal activity (*Bargmann and Marder, 2013*; *Marder, 2012*). Because of many nonlinear interactions among the components of even simple neuronal networks, detailed models are not necessarily conceptually revealing (*Selverston, 1980*) and are computationally costly (*Izhikevich, 2003*; *Markram, 2006*). Finally, bottom-up approaches typically assume that the microscopic parameters measured in a typical experiment such as neuronal connectivity or biophysics of individual neurons and synapses must be tuned to specific values in order to assure proper functioning of the brain. Variations around these values are typically seen as noise. Thus, microscopic parameters are routinely averaged across iterations of the same experiment and across individuals. Yet, biophysically realistic simulations of even simple neuronal networks in crustaceans (*Prinz et al., 2004*) show that the relationship between the microscopic parameters and global behavior of the network is highly degenerate. Many disparate microscopic configurations lead to almost indistinguishable macroscopic behavior. Because of non-linearities, however, averaging microscopic parameters disrupts the global behavior of the system (*Golowasch et al., 2002*). Therefore, in order to adequately constrain a realistic model of a neuronal network, many microscopic parameters need to be simultaneously measured in the same animal (*Golowasch et al., 2002*). Yet, such a detailed model is not guaranteed to be generalizable across individuals. Thus, while on the one hand there is a desire to create sufficiently realistic models, it is likely that ultimately these bottom-up approaches need to be combined with more abstract phenomenological models of neuronal dynamics. Here, we describe a general methodology capable of extracting neuronal dynamics from neuronal imaging in nematode *C. elegans* (*Brennan and Proekt, 2017*). To demonstrate the power of this approach we show that our model is capable of predicting future motor commands on a cycle-by-cycle basis and is valid across multiple individual *C. elegans* despite consistent inter-individual differences in neuronal activation.

Locomotion of *C. elegans* is one of the very few biological systems where experimental measurements of brain activity and behavior can be performed with sufficient granularity for developing and testing a quantitative model of brain dynamics at a behaviorally relevant scale. All 302 neurons (*White et al., 1986*) in *C. elegans* and all their connections are known (*Izquierdo and Beer, 2013*; *Bargmann and Marder, 2013*; *Varshney et al., 2011*). Simultaneous recordings of the majority of the neurons in the brain (head ganglia) of *C. elegans* have been performed in vivo using calcium imaging (*Kato et al., 2015*; *Nguyen et al., 2016*; *Prevedel et al., 2014*; *Tian et al., 2009*) (*Figure 1A*, Materials and methods). The graded activity of most *C. elegans* neurons (see *Liu et al., 2018*, however) make them better suited for calcium imaging compared to vertebrate nervous systems in which the utility of calcium imaging is limited by the slow speed of calcium indicators relative to the temporal precision of spike trains (*Rad et al., 2017*). Biomechanics of locomotion of *C. elegans* are well-described by just a few movement modes (*Stephens et al., 2008*) suggesting that the dynamics of the nervous system that controls locomotion are likely to be simple enough to be inferred from relatively short recordings of neuronal activity. Locomotor behaviors fall into well-characterized individual distinct stereotyped behavioral subtypes (*Kato et al., 2015*; *Li et al., 2014*; *Luo et al., 2014*; *Larsch et al., 2013*) (*Figure 1B*). The final fundamental advantage of *C. elegans* as a model organism is that neurons can be individually identified in different genetically identical animals (*Kato et al., 2015*). Thus, *C. elegans* is an ideal model system for the proof of principle that a model of neuronal dynamics can be constructed on the basis of imaging of neuronal activity.

## Results

Approximately 100 neurons in the head ganglia were recorded simultaneously in each of five animals (*Figure 1A*) immobilized in a microfluidic chamber (*Kato et al., 2015*). Our model is built upon this data set. Using imaging of a limited subset of neurons in freely moving *C. elegans*, *Kato et al. (2015)* verify that activation of some individual neurons is closely associated with parameters of locomotion. Thus, neuronal activity in the immobilized animal has been interpreted as motor commands that signal locomotor behaviors. *Kato et al. (2015)* used activation of individual neurons to assign a fictive locomotor behavior to each point in the observed time series of neuronal activation in the immobilized animals. Throughout this work, we used the behavioral states assigned by Kato et al.

Our objective here is to quantitatively model the sequences of such motor commands. Kato et al. used principal component analysis (PCA) to reveal stereotyped loops in the neuronal activity (*Figure 1—figure supplement 1*). This methodology allows for analysis of the relationship between neuronal population activity and behavior in each individual *C. elegans* (*Kato et al., 2015*). In contrast, we attempt to develop an analytical method that allows for the quantification and prediction of motor commands across multiple individuals. The first step in making this transition is to find a common set of neurons experimentally identified in all animals. Even in the simple nervous system of *C. elegans* not every neuron can be reliably and uniquely identified. Indeed, only 15 neurons were consistently and unequivocally identified in each individual *C. elegans* (top 15 rows in *Figure 1A*). Unfortunately, methods successfully applied by *Kato et al. (2015)* to ~100 neurons in each individual fail to reveal meaningful structure when applied to the common 15 neuron subset across individuals (*Figure 1—video 1*).

There are two putative classes of reasons for this failure. First, it is likely that relevant information is lost when the number of neurons is reduced. Information loss is bound to be more significant in complex organisms whose nervous systems contain orders of magnitude more neurons than *C. elegans*. This information loss could be potentially mitigated by developing novel experimental approaches. However, we will show that uniquely identified neurons in *C. elegans* exhibit consistent statistical differences in their patterns of activation across animals. This observation necessitates the development of new analytical techniques capable of extracting global neuronal dynamics on the basis of variable activation of a limited subset of neurons. In what follows we will first demonstrate that our technique allows for the efficient and accurate simulation of *C. elegans* neuronal activity. We will then show that simulations of neuronal dynamics can also be used to predict behavioral switches up to 30 s before they occur in a different experimental cohort of animals. Finally, we will demonstrate that these predictions are possible because global dynamics of the *C. elegans* nervous system are conserved despite consistent differences in activation of individual neurons.

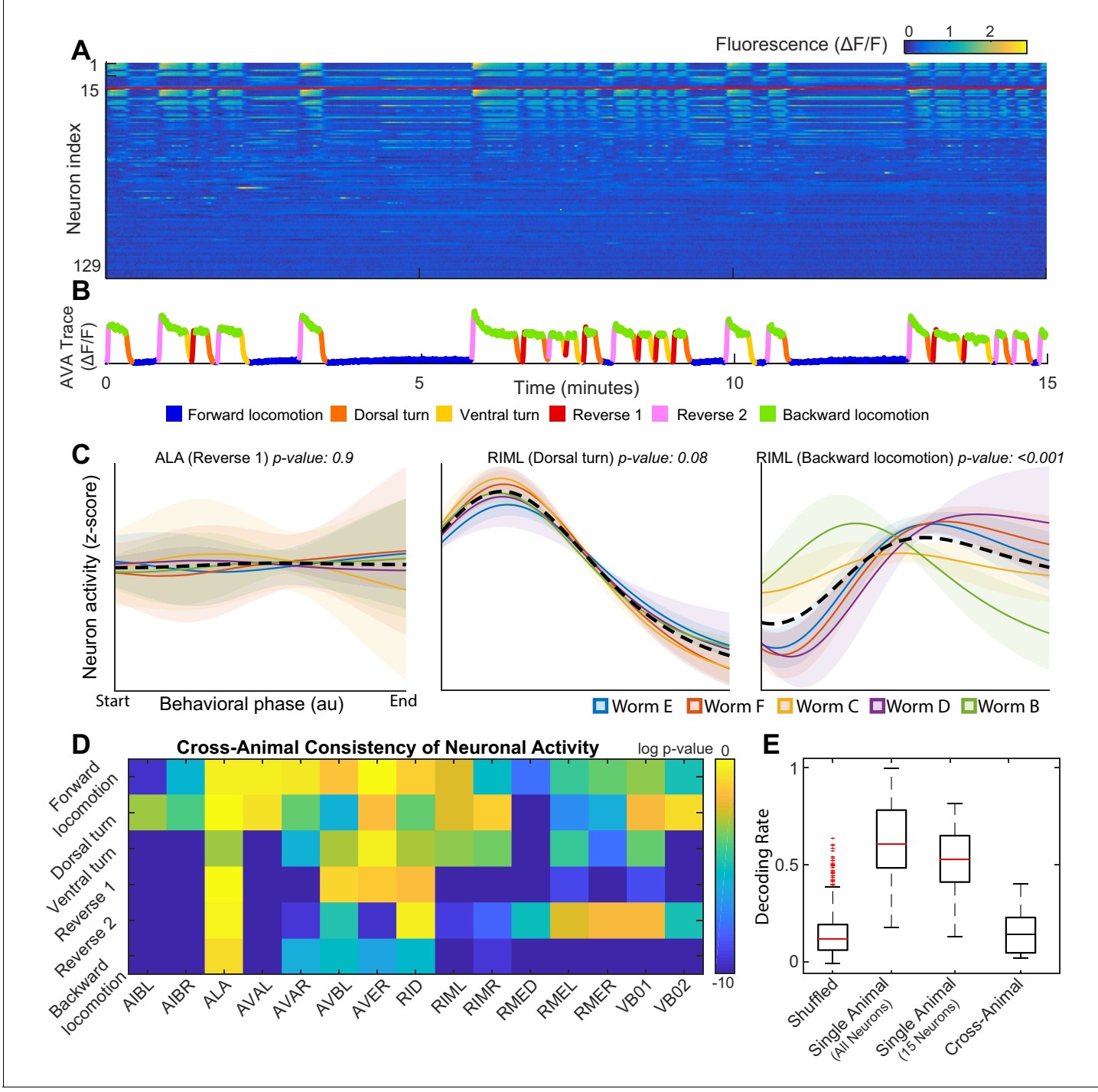

**Figure 1.** Neuronal activity is consistently different in different individuals. (**A**) Calcium signals ($\Delta F/F$) recorded in one animal for ~15 min by *Kato et al. (2015)*. Each row represents a single neuron. The top 15 rows (above the red line) correspond to neurons unambiguously identified in all animals (shared neurons). (**B**) Trace of the AVA neuron colored by behavioral state as defined by *Kato et al. (2015)*. (**C**) Neuronal activity of representative neurons plotted as a function of behavioral phase (Materials and methods) in a single behavior grouped by animal. Colored solid lines show mean activity for each animal. Shaded regions show 95% confidence intervals. Mean and confidence intervals are computed across multiple cycles of the same behavior in each animal. The dashed black line shows the mean across all cycles of the behavior in all animals. The cross-animal mean for ALA and RIML (Dorsal turn) is a good approximation of activity in each animal individually. In contrast, the cross-animal mean of RIML (Backward locomotion) does not represent any individual. Thus, activity of RIML is consistently different among individuals during backward locomotion. (**D**) Probabilities that neuronal activity from different individuals was drawn from the same distribution (Materials and methods) computed for each neuron in each locomotor behavior. Activity of most neurons differs consistently among individuals in at least one locomotor behavior. (**E**) Attempts to decode the onset of

*Figure 1 continued on next page*

*Figure 1 continued*

backwards locomotion are successful within each animal individually using either all neurons or the 15 shared neurons. This confirms that neuronal activation is stereotyped in each animal. Yet decoding fails across animals. Probability of decoding onset of backwards locomotion in one animal on the basis of neuronal activity averaged across other four animals is indistinguishable from chance. Thus, averaging neuronal activity across individuals disrupts behaviorally relevant information. Box plots show distribution of decoding rates bootstrapped across animals and bi-partitions of the data into training and validation datasets (Materials and methods).

DOI: https://doi.org/10.7554/eLife.46814.003

The following video and figure supplement are available for figure 1:

**Figure supplement 1.** Projection of calcium imaging data onto Temporal PCs.
DOI: https://doi.org/10.7554/eLife.46814.004
**Figure 1—video 1.** Projection of calcium imaging data onto shared TPCAs.
DOI: https://doi.org/10.7554/eLife.46814.005

## Variable activation of identified neurons in *C. elegans*

One plausible explanation of variability in neuronal activity is that a particular neuron is irrelevant for a specific behavior and therefore its activity is not adequately constrained. An example of this type of variability is ALA – a neuron involved in quiescence regulation and mechanosensation (*Van Buskirk and Sternberg, 2007*; *Sanders et al., 2013*; *Hill et al., 2014*; *Nelson et al., 2014*). Since experiments analyzed herein were performed in immobilized worms and no quiescence was observed, as expected, ALA activation is quite variable from one cycle of reversal to the other in each individual animal. Note, however, that there are no statistically significant differences between ALA activity during reversals across different individuals (*p-value ≈ 0.9*, Materials and methods) (*Figure 1C*). As a result, neuronal activity averaged across animals at each phase of behavior is representative of neuronal activity observed in each animal individually. In contrast variability of activation of RIML – a command neuron known to activate AVA which, in turn, elicits backwards locomotion (*Guo et al., 2009*) – is paradigmatically distinct. During backwards locomotion, RIML activation differs significantly between animals (p-value < 0.001, Materials and methods). These differences are not simply random noise superimposed onto a common activation template. As a result, averaging RIML activity across animals during backward locomotion yields a pattern of activity that does not resemble that observed in any one of the individual *C. elegans*. Yet, during a different behavior – dorsal turn – RIML activation is consistent across individuals (p-value ≈ 0.1), Materials and methods. This makes it unlikely that the observed differences in RIML activation during backwards locomotion are an artifact of neuron misidentification.

Consistent differences in activity of individual identified neurons between genetically identical animals performing the same behavior are not unique to RIML. To show this, we quantify inter-individual differences in activity of each neuron during each locomotor behavior (*Figure 1D*). The p-values in *Figure 1D* reflect the probabilities that activation of a particular neuron is consistent among individuals. For most neurons involved in locomotion activity differs from animal to animal during execution of at least one type of locomotor behavior. Many neurons can be consistently activated in one locomotor behavior but be highly inconsistent among individuals in another type of locomotion. Only three neurons were consistent in all behaviors. One of these neurons (ALA) is not known to play a direct role in locomotion beyond quiescence (*Van Buskirk and Sternberg, 2007*; *Sanders et al., 2013*; *Hill et al., 2014*; *Nelson et al., 2014*) not observed in this dataset. Consistent with this observation, ALA did not exhibit any appreciable activation during any locomotor behavior. AVB and RID were the only locomotion-associated (*White et al., 1986*; *Lim et al., 2016*) neurons whose activity failed to exhibit statistically significant differences among individual animals in any locomotor behavior. These inter-individual differences in neuronal activation is the primary reason why principal component analysis performed on neuronal activity in each individual successfully reveals cycles in neuronal dynamics (*Kato et al., 2015*) but attempts at projecting data from all individuals onto a common set of principal components fails to reveal any meaningful structure (*Figure 1—video 1*).

To further illustrate the consistent differences in neuronal activation among individuals, we attempted to decode the behavioral state on the basis of neuronal activity. Half of all instances of backing behavior were used to compute the average activity of each neuron at the onset of backing behavior. Mutual information between this snapshot of neuronal activity and behavioral state

(Materials and methods) was then used as the basis for decoding the other half of backing behaviors either within each animal or across animals. Using this strategy, we reliably decoded the onset of backwards locomotion based on ~100 neurons recorded in each animal individually (p-value < 0.001 relative to shuffle control, Materials and methods). The ability to decode did not degrade appreciably when just 15 neurons identified in each animal were used (p-value ≈ 0.5 within animal - all neurons vs. within animal - 15 neurons, Materials and methods). This limited subset of neurons (~1/20th of the entire nervous system), therefore, still contains most of the essential information about initiation of backwards locomotion and confirms that neuronal activation is consistent in each animal. This is not surprising as the 15-neuron subset contains most of the known command neurons that control the direction of locomotion.

Yet, activity from one animal cannot be used to reliably decode the onset of backing behavior in another animal. When activity from four animals was used to decode the 5th (leave one out) the correct decoding rate was indistinguishable from chance (p-value ≈ 0.3, Materials and methods) (*Figure 1E*). Thus, mutual information between neuronal activity and behavioral state is degraded when neuronal activity is averaged among genetically identical individuals during locomotion in a simple environment. This inter-individual variability is the fundamental reason why simple averaging of activation of individual neurons fails to yield a meaningful model of neuronal dynamics. Although there are potentially many different classifiers that could be built to decode the behavioral state on the basis of neuronal activity, a classifier based on mutual information is a parsimonious strategy that succeeds in decoding behavior in each individual. Thus, it is unlikely that our ability to decode the behavioral state on the basis of neuronal activity will be dramatically improved by a different classification strategy.

Consistent differences in activation of individual neurons do not necessarily imply that global dynamics of the brain are distinct in different individual *C. elegans*. It is possible that distinct activity combinations observed in different individuals give rise to an equivalent behavioral strategy implemented at the level of global brain dynamics. An example of this state dependence of neuronal activity is known in the olfactory system of *C. elegans* (*Gordus et al., 2015*). This degeneracy of neuronal activation complicates analysis of individual microscopic components taken in isolation or averaged across individuals.

## Underlying neuronal dynamics give rise to neuronal activity

There is a fundamental distinction between neuronal activity and neuronal dynamics (*Churchland et al., 2012*; *Salinas and Sejnowski, 2001*). Neuronal dynamics are the laws of motion that govern the temporal evolution (flux) of neuronal activity in the space spanned by the relevant variables (phase space). Thus, rather than focusing on individual neurons, the dynamical systems description is focused on identifying the salient variables that make up the phase space and on the laws of motion that act to move the state of the system along a trajectory in phase space. The observed neuronal activity is governed by the biophysics of individual neurons and synapses (*Seung, 1996*; *Beer, 1995*; *Miller and Selverston, 1982*) as well as activity of other neurons not reliably identified in all experiments. These biophysical processes influence neuronal activity and are in turn influenced by it. Yet, these processes cannot be directly inferred from the observed activation of neurons.

In the appendix, we illustrate a novel method – Asymmetric Diffusion Map Modeling – that allows for the extraction of neuronal dynamics from high-dimensional, noisy and non-linear neuronal activity time series recordings. The final output of this method is a two dimensional approximation of the neuronal dynamics which describes the time evolution of the system as a flux along distinct loops in phase space.

One fundamental advantage of having an approximation of neuronal dynamics is that neuronal activity in *C. elegans* can be efficiently simulated (*Figure 2A*). The validity of the simulated dynamics can then be explicitly tested by comparing these newly simulated traces of neuronal activity to those experimentally observed in *C. elegans*. This simulation is first performed in the phase space. As the system evolves in phase space it traces out neuronal activation (*Figure 2A*) (Materials and methods and Appendix). Note, that the simulated neuronal activity does not merely recapitulate experimental observations but rather yields *new* neuronal activity traces. These simulated activity traces are in good qualitative agreement with experimental observations. Both the observed and the simulated traces exhibit abrupt coordinated transitions between levels of activity of multiple neurons. Further

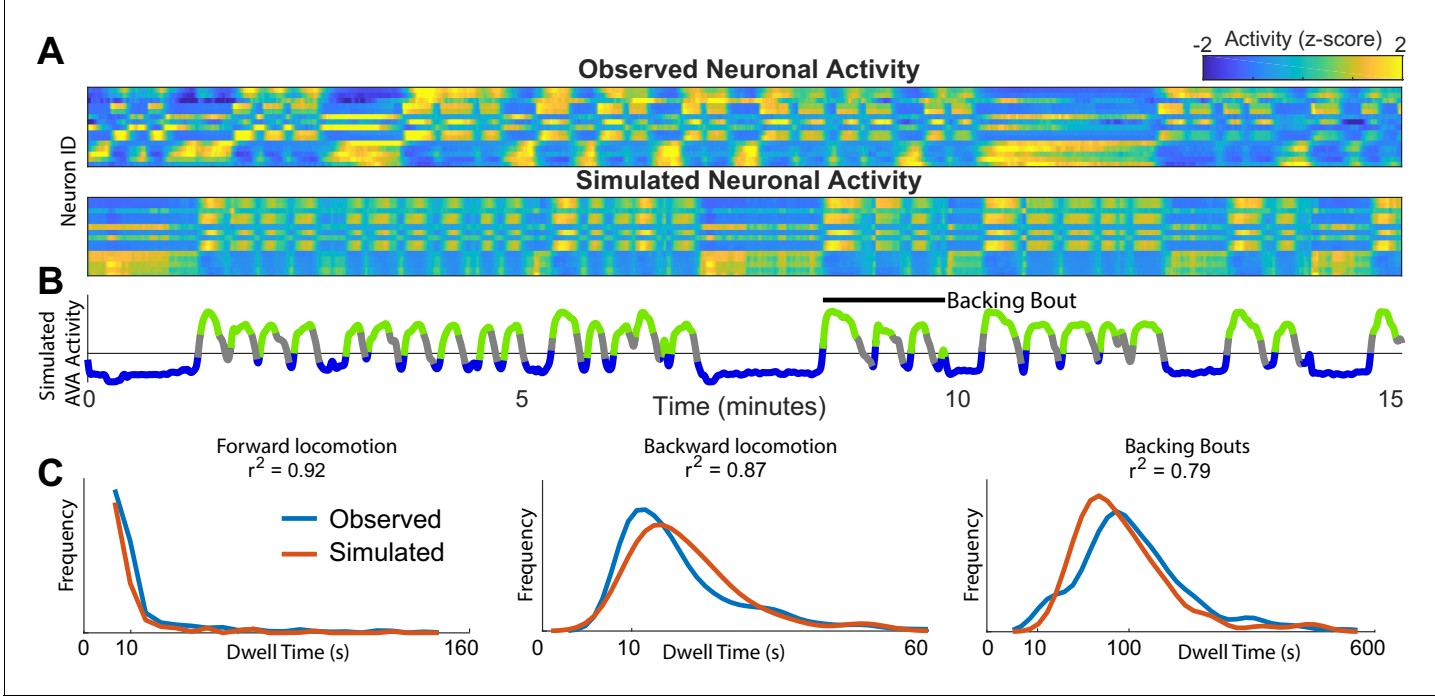

**Figure 2.** Simulations of the dynamics faithfully reproduce neuronal activity and behavioral statistics. (A) Experimentally observed (top) and simulated (bottom) activity of 15 shared neurons plotted as in *Figure 1A*. (B) Trace of the simulated AVA neuron colored according to the inferred behavioral state. In contrast to *Figure 1*, color here expresses neuronal activity as *z-score* computed for each neuron individually. Behavioral states are assigned based on experimentally observed distribution of behaviors for each point in phase space (blue → forward locomotion; green → backward locomotion) (Materials and methods). Both A and B are plotted on the same time-axes. Because of under-sampling, transitions between forward and backward locomotions are left unassigned (gray). (C) Dwell time distributions for forward locomotion, backward locomotion and backing bouts (blue → experimentally observed; orange → simulated). Backing bouts were defined as repeated episodes of backing behavior separated by short forward locomotion states lasting at most 30 frames ( 10 s) (e.g. black line in B). Although the manifold is constructed on the basis of transition probabilities between states separated by <1 s., the manifold successfully predicts statistics of behaviors over 100 s.

DOI: https://doi.org/10.7554/eLife.46814.006

The following figure supplements are available for figure 2:

**Figure supplement 1.** Neuronal activity trace spectral residues.
DOI: https://doi.org/10.7554/eLife.46814.007

**Figure supplement 2.** Auto correlations and auto mutual information inform expected delay time scales.
DOI: https://doi.org/10.7554/eLife.46814.008

**Figure supplement 3.** Effect of delay embedding parameters on model accuracy.
DOI: https://doi.org/10.7554/eLife.46814.009

**Figure supplement 4.** Model can be constructed on the basis of a single neuron.
DOI: https://doi.org/10.7554/eLife.46814.010

note that the correlations in activation across neurons are preserved. Finally, note that the activity of the simulated AVA neuron (*Figure 2B*) exhibits bouts of activations interspersed with prolonged periods of inactivity corresponding to backward and forward locomotion respectively. These bouts are in good qualitative agreement with the experimental observations. The first instance of backward locomotion in a bout is distinct from subsequent instances. It is associated with stronger activation of the AVA neuron (*Figure 1B*). Remarkably, transient activation is also a salient feature of the simulated AVA during the first instance of backing behavior in a bout (*Figure 2B*). Because there is an element of stochasticity in the neuronal dynamics, the total number of instances and durations of locomotor behaviors are variable both in the experimentally observed and simulated neurons. To quantitatively compare the simulated and observed neuronal activation, we computed the spectra of each of the 15 neurons identified across all individuals to the spectra of simulated neurons (Materials and methods). With the exception of the very low frequencies (<0.05 Hz) most strongly

affected by the finite dataset effects, the spectra of all simulated neurons are statistically indistinguishable from experimentally observed neuronal activity (*Figure 2—figure supplement 1*).

To determine whether the model of neuronal dynamics reproduces behavioral statistics, we assigned each time point in a simulation a behavioral state. This was accomplished by sampling the empirically derived distribution of behaviors at each point in phase space. The experimentally observed and de novo simulated distributions of dwell times in different behavioral states are in excellent agreement (*Figure 2B*). Note that the simulations reproduce not just the time scale of individual behaviors (forward and backward locomotion) but also sequences of behaviors that we refer to as backing bouts. This is remarkable because the model of the dynamics was constructed by estimating probability of transition between two states on the time scale of one time step dictated by data acquisition and GCAMP kinetics (~ 1/3 of a second). Yet, the simulation reproduces the dynamics on the time scale longer than 100 s. Note that PCA previously applied to neuronal activity (*Kato et al., 2015*) does not directly yield a quantitative model that can be used to simulate *new* neuronal activity. Thus, inter-individual variability aside, PCA in and of itself does not yield any quantitative predictions concerning neuronal dynamics.

## Simulations of neuronal dynamics predict behavioral switches

Based on the observations of abrupt stereotyped transitions in activity of many neurons (e.g. *Figure 1A*) and dwell times of locomotor behaviors, it has been argued that switching between different modes of locomotion in *C. elegans* is stochastic (*Roberts et al., 2016*; *Srivastava et al., 2009*). If so, then timing of behavioral transitions on each individual cycle of behavior should be unpredictable and the entirety of information concerning behavioral switching should be contained in the dwell time distributions.

Thus, the most compelling test of the neuronal dynamics model is the ability to predict future abrupt changes in neuronal activation that signal switches in locomotor behavior solely on the basis of initial position in phase space. To test this prediction, we make use of a new dataset of calcium imaging in *C. elegans* from *Nichols et al. (2017)* (Materials and methods). We restricted our analysis to the prelethargus N2 animals (n = 11) that were subjected to similar experimental conditions and imaging to those from *Kato et al. (2015)* dataset. Critically, no data from the Nichols et al. dataset was used for the construction of the model. Animals in the dataset (*Nichols et al., 2017*) shared between 8 and 13 neurons with the neurons recorded by *Kato et al. (2015)* on the basis of which the neuronal dynamics model was constructed (Materials and methods).

Simulations started from several initial positions (phase bins) associated with backwards locomotion were used to estimate the expected distribution of times to the start of forward locomotion (*Figure 3A*, orange) for each phase bin. To compare these predictions to the experimental observations, we identify all points in the validation dataset from Nichols et al. that pass through the same phase bins and note the experimentally observed time until the start of forward locomotion signaled by abrupt change in AVA activity (*Figure 3A*, blue). For most phase bins, the expected time of simulated behavioral switch was indistinguishable from experimentally observed switch in motor command. In contrast, the predictions made by the null model based solely on behavioral dwell time distributions deviate significantly from the timing of observed transitions.

To quantify the success of the predictions, we compute the correlation between simulated time to initiation of forward locomotion and that observed by Nichols et al. for each phase bin (*Figure 3B*). Consistent with observations in (*Figure 3A*) simulation-based predictions (filled circles) were strongly correlated with observed timing of behavioral transitions (correlation coefficient 0.74) (*Figure 3B*). In contrast, predictions based solely on the dwell time distributions were less well correlated (p < 0.0001) with experimental observations. Further, note that the dispersion around the best fit line is smaller for the simulation-based than for dwell-time based predictions. Thus, dynamics-based predictions are more precise and accurate than those based on behavioral statistics alone. Because definition of behavioral states relies heavily on observed activity of the AVA neuron, we sought to determine whether including AVA critically affects the results. We removed AVA from the *Kato et al. (2015)* dataset used for model construction and the *Nichols et al. (2017)* dataset used for model validation. Even in the absence of the AVA, manifold predictions correlated strongly with the observed time of behavioral transitions (Slope 0.9; $R^2$ 0.8) and outperformed predictions based solely on dwell time distribution *Figure 3—figure supplement 1*. Therefore, our modeling approach reveals a strong contribution of deterministic dynamics to abrupt changes in locomotor direction in

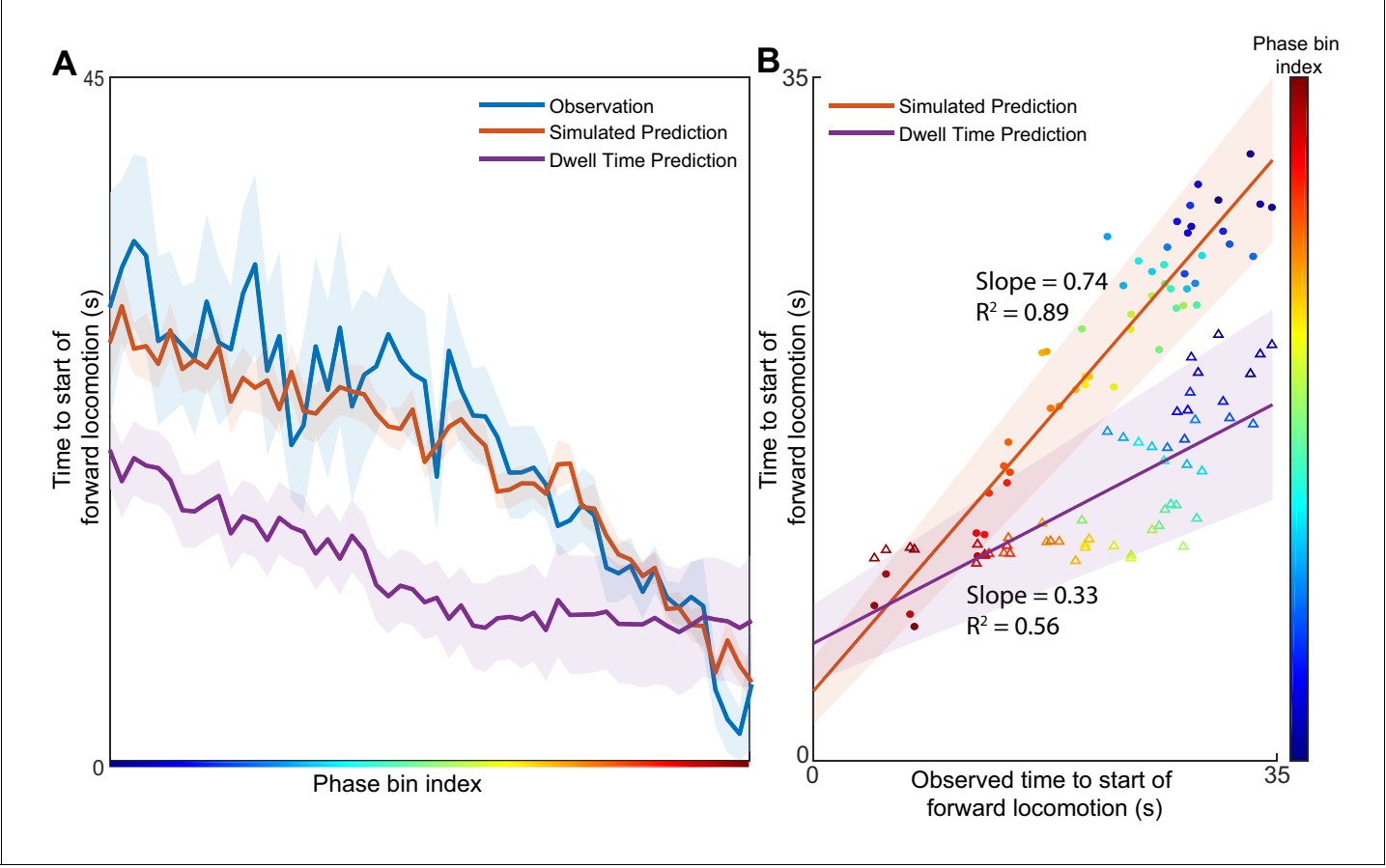

**Figure 3.** Position in the phase space determines the expected time of future behavioral transitions. We explore the region of phase space where most instances of motor commands for backward locomotion are terminated and give rise to commands for forward locomotion. (**A**) The distribution of times to start of forward locomotion (orange line → mean, orange shading → 95% confidence intervals) is simulated as a function of starting phase. Observed distribution of times to start of forward locomotion in 11 new animals from Nichols et al. (blue line → mean, blue shading → 95% confidence intervals) are calculated for each phase bin in the same region. For each phase bin we also computed the average time since onset of backward locomotion. We then used this average time and the dwell time distribution of backward locomotion to calculate the expected time of termination of backward locomotion (purple line → mean, purple shading → 95% confidence intervals) (Materials and methods). This null hypothesis is grossly inaccurate. (**B**) Times to start of forward locomotion for both the simulation (orange line → mean, orange shading → 95% confidence intervals, filled circles), and the null model built on dwell times (purple line → mean, purple shading → 95% confidence intervals, empty triangles) are plotted against the time to start of forward locomotion command of the newly observed Nichols et al. animals. Individual points show the average time to start of forward locomotion for each individual phase bin (point color). The simulation (orange) is a good predictor of the newly observed data. In contrast, the null model based on dwell times (purple) consistently underestimates the time until initiation of forward locomotion (p < 0.0001).

DOI: https://doi.org/10.7554/eLife.46814.011

The following figure supplement is available for figure 3:

**Figure supplement 1.** Predictions of expected time to switch of behavior are still valid without AVA.

DOI: https://doi.org/10.7554/eLife.46814.012

*C. elegans*. These predictions do not depend strongly on activity of AVA–the command neuron for backward locomotion. It should be noted, however, that by construction the Asymmetrical Diffusion Map Method is a stochastic model. Thus, in addition to the deterministic cyclic fluxes, stochastic forces also contribute to the observed neuronal activity. Remarkably, the method reveals that the transition probability between neuronal activity patterns is a function of the macroscopic variables such as phase of the cyclic flux.

Knowing the initial conditions is sufficient to predict the expected time of transitions between different modes of locomotion 30 s before they are experimentally observed (*Figure 3*). Remarkably, these predictions are valid across individuals observed years apart. Therefore, neuronal dynamics model can be applied universally across individuals despite significant inter-individual differences in

neuronal activation and undersampling of neuronal activity. Although it is likely that the simulation-based predictions could be improved with addition of more neurons, the fact that the animals in the validation dataset shared as few as eight neurons with the original data suggests that using our methodology one can uncover macroscopic dynamics even when only a small subset of the nervous system can be recorded and unequivocally identified.

In principle, our methodology (Materials and methods and Appendix) could be used to uncover system dynamics from activity of any single component of a tightly coupled system (*Harnack et al., 2017*). Thus, we attempted to reconstruct dynamics of *C. elegans* nervous system on the basis of activity of a single neuron. We used a single neuron from the *Kato et al. (2015)* for model construction. The quality of predictions was assessed using dwell time statistics (Materials and methods) *Figure 2—figure supplement 4*. The quality of predictions varied substantially between neurons. Models built on some neurons involved in backwards locomotion (e.g. AVAL, AVAR, AVER, and RIML) yielded predictions comparable to those obtained for a set of 15 neurons. In contrast, neurons that play limited role in locomotion such as the ALA were not predictive. Interestingly, although RIML is known to play a role in backward locomotion, its activity varied significantly among individual animals (*Figure 1C*) during backwards locomotion. Nevertheless, models based solely on RIML were ~75% as informative as models built upon the entire 15 neuron set. Thus, at least in the simple nervous system of *C. elegans* a predictive model can be constructed on the basis of a single experimentally observed neuron as long as activation of this neuron is tightly coupled to the network that mediates the observed behaviors.

## Macroscopic dynamics are conserved among animals

The ability to simulate neuronal activity, behavioral dwell-time statistics, and even predict timing of individual behavioral transitions implies that trajectories traced by the state of the brain as it evolves in phase space are remarkably conserved among individuals. If the dynamics that give rise to neuronal activity were purely deterministic, then such trajectories would never cross (*Sugihara et al., 2012*; *Strogatz, 2014*). However, any experimental system is bound to have noise due to both measurement error and stochastic processes that affect the trajectories traversed in phase space. Noise inevitably causes trajectories to tangle. Nevertheless, in the limit of low noise (Materials and methods), trajectories will form bundles in phase space. A collection of such trajectory bundles is referred to as the manifold.

To determine whether the manifolds are conserved among individuals, we applied the manifold reconstruction method (Materials and methods and Appendix) to neuronal activity of *C. elegans*. The manifold in *Figure 4A* was constructed on the basis of all 107 neurons recorded in one animal. This illustrates that our methodology is able to reconstruct the global dynamics in the limit of relatively large fraction (~ 1/3) of all neurons (*Figure 4—figure supplement 1*) and can be applied to time series consisting of at least 100 neurons. In the *C. elegans* nervous system, the phase space (Materials and methods) is too high dimensional to be shown graphically in its entirety. Nevertheless, trajectories spanned by a broad class of noisy dynamical systems (*Wang et al., 2008*) will form loops – a low-dimensional object in the high-dimensional phase space. Thus, a position of the system can be approximated just by two parameters: the identity of the loop $\alpha$ and the phase along it $\theta$. Identifying these variables from neuronal activity (Materials and methods) allows us to project neuronal activity averaged with respect to $\theta$ and $\alpha$ onto the first three principal components. This coordinate system which we refer to as DPCA plays no role in simulating neuronal dynamics and is used purely for visualization purposes (*Figure 4—video 1*). The width of the manifold represents the density of points or, equivalently, decreases in phase velocity $d\theta/dt$. The direction of phase velocity is shown by arrows. For instance, in the region associated with forward locomotion (blue) phase velocity is relatively small. Thus, transit through this region of phase space is dominated by stochastic processes. In contrast, reversal behaviors (red and purple) are associated with high $d\theta/dt$. Therefore, duration of reversals has a characteristic time scale dominated by phase velocity. The sequence of behaviors is dictated by the arrangement of different locomotor behaviors along the phase of the manifold. The distribution of locomotor behaviors as a function of position in the manifold is shown by color. The final color of the manifold is a blend of the colors for each behavior according to their prevalence. Note that although behavioral assignments were not used in the construction of the manifold (Materials and methods), most regions of the manifold are associated with just one type of locomotor command. In other words, different locomotor commands are localized to different regions in

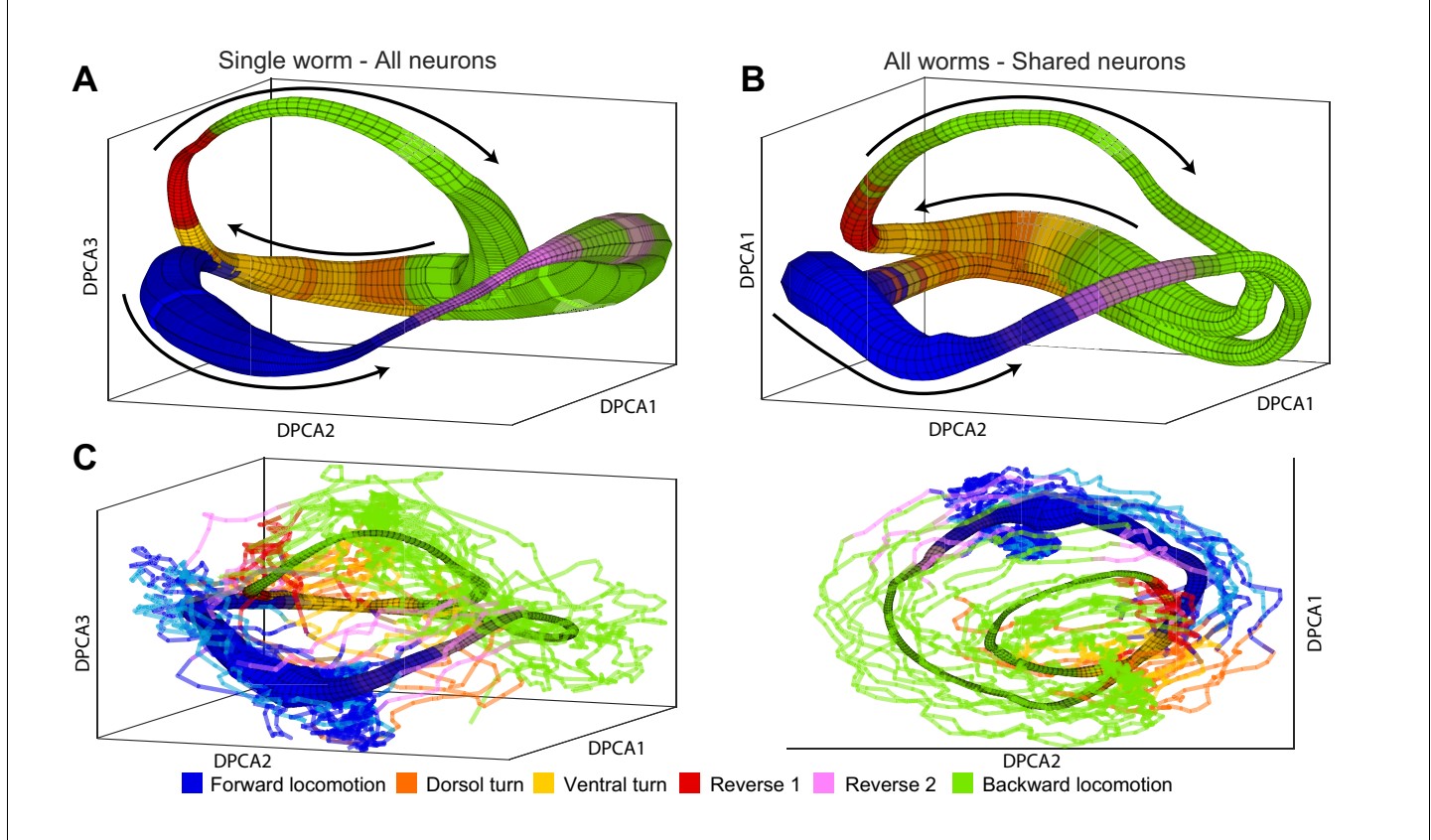

**Figure 4.** Manifolds are invariant across animals. (**A**) Manifold constructed for a single animal using all 107 neurons in the recording. Manifold is color coded according to locomotor behavior assigned to each snapshot of neuronal activity by Kato et al. The prevalence of each behavior in each phase bin $\Delta\theta$ is encoded in the opacity of the color for each behavior (colors corresponding to each behavior are shown at bottom of the figure). Black arrows represent the direction of flux $d\theta/dt$ around the manifold. DPCA1-3 are the first three principle components of the neuronal activity averaged with respect to $\theta$ and $\alpha$. (**B**) Manifold constructed using the data from all animals using the 15 shared neurons as common basis. Coloring, projection and arrows are the same as in (**A**). (**C**) The data from one animal (lines) is projected onto the manifold constructed from four other animals (surface). The trajectories of the left out animal are well-approximated by the manifold built entirely on the basis of other animals. Behavioral states of the left out animal align well with the distribution of motor commands along the phase of the manifold. This allows us to decode the motor command reliably on the basis of position in manifold space $(\theta, \alpha)$ across animals. The same manifold and data is shown in two different projections in order to emphasize the similarities between manifold and data.

DOI: https://doi.org/10.7554/eLife.46814.013

The following video and figure supplements are available for figure 4:

**Figure supplement 1.** Projection of calcium imaging data onto DPCAs.
DOI: https://doi.org/10.7554/eLife.46814.014
**Figure supplement 2.** Asymmetric diffusion matrix.
DOI: https://doi.org/10.7554/eLife.46814.015
**Figure 4—video 1.** Projection of calcium imaging data onto shared DPCAs.
DOI: https://doi.org/10.7554/eLife.46814.016

the phase space. While the two trajectory loops are well separated, the system is quite deterministic. When the two loops pass near each other, conversely, the future state of the system is dominated by stochastic processes.

Several lines of evidence converge on the fact that, unlike activity of individual neurons, the phase space $(\theta, \alpha)$ is universal across animals. The manifold in *Figure 4B* was constructed on the basis of activity from all five animals using only 15 neurons identified in each animal. In contrast to averaging neuronal activity by applying PCA (*Figure 1—figure supplement 1*) reconstruction of neuronal dynamics is possible even when only 15 neurons (~5%) are consistently identified in each individual. This is especially remarkable given the inter-individual differences in activation of the common

neuronal subset. The structure of the manifold constructed on the basis of 15 neurons across individuals is nearly identical to the manifold constructed on the basis of 107 neurons in a single animal (*Figure 4A*).

Position in phase space $(\theta, \alpha)$ preserves behavioral information across animals. As a result, the assigned behavioral state can be correctly decoded 83% of the time solely on the basis of position along the manifold in *Figure 4B*. This is the median successful decoding probability computed across across all locations in the manifold binned into 426 bins (total of 15405 predictions in all five animals, $\chi^2$ 19.8, p-value $5.5 \times 10^{-4}$). Note that because of limited temporal precision with which behavioral states can be experimentally assigned, some uncertainty about the behavioral state is expected especially around the times of behavioral transitions. To further strengthen the argument for universality of the global dynamics, we constructed a manifold based on the data from four out of the five animals in the *Kato et al. (2015)* dataset. We then used the manifold to project the neuronal activity from the fifth animal *not used* for manifold construction onto the manifold space (*Figure 4C*). Behavioral states of the left out animal align well with the distribution of behavioral states along the manifold. Correct behavioral state assignment in the excluded animal can be decoded 81% of time (median correct decoding probability across all 426 manifold bins and all five animals left out in turn). The difference in the median correct prediction probability based on the all worm manifold and the leave one out manifold (*Figure 4C*) is not statistically significant ($\chi^2$ 3.8, p-value 0.44). The probability of obtaining this quality of decoding by chance is $p = 0.0014$ ($\chi^2$ 17.7).

Thus, averaging neuronal activity with respect to its position in phase space, rather than across individual neurons, preserves most of the behavioral information and can be universally applied across individuals even when only ~5% of neurons are uniquely identified. This conserved shape of the manifold in the phase space is what allows the predictions of timing of switching of motor commands across different animals. Yet, the salient variables that span the phase space are not directly apparent from recordings of individual neurons even when most locomotor control circuitry is recorded in a simple environment.

## Discussion

Here, we developed a method for extracting salient dynamical features from complex, multivariate, nonlinear, and noisy time series. We apply this method to neuronal imaging in *C. elegans* to demonstrate its success in simulating activity of the nervous system and predicting switches between different motor commands. The manifold in *C. elegans* nervous system is composed of two loops. While the system is in either one of the loops, its fate is largely predictable. Yet, in the neighborhood where the loops merge, the behavior cannot be clearly predicted and stochastic forces play a stronger role. This leads us to hypothesize that the region where the two loops merge is a decision point where the nervous system is most susceptible to noise and/or sensory inputs (*Gordus et al., 2015*). The manifold shape is conserved among individuals and initial position in the manifold space is sufficient to predict future switches in motor commands. This suggests that the macroscopic variables such as loop identity and phase along it express behaviorally relevant information.

Intriguingly, we find that even in genetically identical organisms consistent differences in neuronal activity associated with motor commands are the norm. This striking observation is not without precedent. Hodgkin-Huxley models of conductances measured in individual AB neurons in crustacean stomatogastric ganglion exhibit bursting akin to the biological neuron. However, averaging conductance measurements across AB neurons in different individuals yields models that fail to burst (*Golowasch et al., 2002*). Virtually indistinguishable network activity patterns can arise from distinct biophysical mechanisms (*Prinz et al., 2004*; *Chiel et al., 1999*; *Beer et al., 1999*). This suggests that differences between individual AB neurons (*Goldman et al., 2001*; *Prinz et al., 2004*) or individual *C. elegans* are not simply random deviations from a common template that can be averaged away at the microscopic level. This nontrivial inter-subject variability is the fundamental difficulty impeding the construction of biophysically-realistic models of even simple nervous systems. In order to sufficiently constrain such models many parameters have to be simultaneously measured in each individual. This is not currently possible even in the simplest neuronal networks. Even more troubling is the observation that such detailed models may not be generalizable between highly similar individuals. Therefore, a more abstract phenomenological approach to modeling neuronal dynamics will be helpful for understanding circuit-level function.

We hypothesize that the nontrivial degeneracy between microscopic biophysical processes and circuit-level dynamics arises because evolutionary selection operates at the macroscopic level of organismal behavior (*Lässig and Valleriani, 2008*) embodied by the global dynamics of the brain. Thus, there is no explicit selective pressure for each individual to produce identical neuronal activation during behavior. Nor is there an explicit pressure for an AB neuron to express a particular number of each of the ion channels on its surface. All that is required is that the overall system gives rise to an adaptive behavioral strategy (*Beer, 2000*). Although undoubtedly there are important constraints imposed by the biomechanics of the animal, the connectome, and other variables, any microscopic solution that gives rise to the appropriate macroscopic dynamics yields the same behavioral strategy. This is equivalent to David Marr's (*Marr, 1982*; *Frégnac, 2017*) proposal that the biophysical details of neuronal circuits are constrained by the computation implemented by the circuit as a whole, rather than the traditional bottom-up approach (*Markram, 2006*; *Markram et al., 2015*) which assumes the opposite. Thus, one should not necessarily expect a detailed model of the nervous system to be equally valid for different, seemingly identical, individuals.

Our methodology can be used to construct a model of macroscopic dynamics despite consistent differences in neuronal activation in different individuals. To appreciate the full computational significance of macroscopic dynamics, future work can apply similar methodology to determine how these dynamics are altered by interaction with the environment (*Clark, 1998*; *Beer, 2000*; *Linderman et al., 2019*). The model in this work was constructed on the basis of immobilized animals. Although *Kato et al. (2015)* established some essential similarities between activation of neurons in the immobilized and freely moving *C. elegans*, there are also important differences (*Nguyen et al., 2016*; *Venkatachalam et al., 2016*; *Scholz et al., 2018*). One important difference is that repeated bouts of backing behavior are not observed in the freely moving animal. Yet, neurons associated with backing behavior (e.g. RIM, AVA, AVE, AIB) were consistently activated during backing in freely moving animals and during fictive locomotion in the immobilized worms. The manifold of *C. elegans* dynamics consists of two loops dominated by forward and backward locomotion. The decrease frequency of backward behavior in the freely moving *C. elegans*, therefore, may correspond to the decreased probability of entering the backward locomotion loop rather than a fundamental differences in the shape of the manifold. Decoupling the motor commands from the behavioral output can prolong the duration of backing behaviors as evidenced by prolonged depolarization of RIM in the immobilized state. This could correspond to the decrease in the phase velocity along the corresponding loop of the manifold. *Kato et al. (2015)* show that silencing the AVA – a command neuron for backward locomotion – eliminates backing behaviors in the freely moving animal. Silencing of the AVA slightly attenuated the activation of RIM and AVE but did not affect the phase relationship between activation of RIM and AVE and other neurons. Thus, although it is possible to uncouple the dynamics of the motor command circuitry from the actual execution of behavior, the macroscopic dynamics remain qualitatively similar. Yet, in general, it is very likely that the manifold shape and properties will depend strongly on the interactions with the environment. Thus, behavioral significance of neuronal dynamics could only be clearly established by reconstructing the neuronal dynamics in animals engaged in their natural behaviors. Nevertheless, our methodology for extracting neuronal dynamics should still apply.

The principal innovation of our methodology is to find loops in non-linear, multivariate and noisy neuronal activity. Oscillations in neuronal activity are well known in nervous systems from leach swimming (*Kristan and Calabrese, 1976*), to stomatogastric ganglia of crustaceans (*Selverston and Moulins, 1985*), to locomotion in primates (*Churchland et al., 2012*) and others. Although oscillations in neuronal activity are expected during rhythmic behaviors, behaviors that are not themselves obviously rhythmic – such as preparation for movement (*Churchland et al., 2010*) or reaching (*Churchland et al., 2012*) – are also associated with rotations in phase space. Thus, we expect that our methodology will be broadly useful for characterizing dynamics in diverse nervous systems.

Several issues need to be considered before applying this methodology to other organisms. The graded potentials of *C. elegans* neurons can be thought of as similar to fluctuations in the firing rate of vertebrate neurons. Yet, it is not always clear whether timing of individual action potentials conveys meaningful information (*Theunissen and Miller, 1995*). In principle, the methodology could be adapted to utilize spike train distances (*Victor and Purpura, 1997*). However, as the number of dimensions of neuronal activity grows, the notions of local neighborhoods become complicated (*Aggarwal et al., 2001*) and may require modifications to the distance measures. Furthermore, the

choice of distance measure and the size of the local neighborhood can effect the coarseness with which neuronal trajectories are combined into the same bundle or split between different bundles of the manifold. Our ability to build a single model that captures the dynamics in different individuals relies on the ability to identify the same neuron in different *C. elegans*. Neuron identification is challenging even in simple systems such as *C elegans* and is generally impossible for complex nervous systems of vertebrates. The fact that the model can be built on a small subset of neurons suggests a possibility that models constructed for different individuals can nevertheless be combined in the manifold space rather than in the space spanned by neuronal activity. In order to accomplish this, future work will need to develop a methodology to robustly compare diffusion maps constructed on the basis of neuronal activity without relying on neuronal identification.

In *C. elegans*, we are able to successfully build a manifold on the basis of ~100 neurons. The effective dimensionality of the data, however, is much smaller. Indeed, we are able to construct a manifold on only 15 neurons and still faithfully simulate the dynamics. Furthermore, the animals in the validation dataset shared as few as eight neurons with the manifold. Nevertheless, the predictions based on the manifold were highly accurate. Because nonlinear dynamical systems are best thought of as wholes rather than a collection of individual components (*Harnack et al., 2017*), the phase space of the nervous system can theoretically be extracted from any individual neuron (*Takens, 1981*). Consistent with this notion, we showed that recording of a single neuron can be used to construct a meaningful model. The reconstruction is only possible, however, when the components of the system are tightly coupled. Only some neurons yielded meaningful predictions in *C elegans*. Thus, recordings from more complex nervous systems may have to first be separated into weakly coupled component parts before the dynamics can be adequately modeled. There is clearly still much work to be done before dynamics of arbitrarily complex and noisy neuronal circuits can be reliably modeled. Nonetheless, our success in modeling the global dynamics of *C. elegans* in a simple environment illustrates the potential power of our method and promises a fruitful new approach to analysis of complex nervous systems.

## Materials and methods

### Non-parametric modeling of global dynamics

Here, we developed a novel method for the extraction of the global dynamics which give rise to observed neuronal activity. We call this method Asymmetric Diffusion Map Modeling. This section will strive to give an overview of the method and a basic intuition as to why it works. A full treatment of the mathematics of the method can be found below. First we will define several distinct representations of the data which the method utilizes. Then, we will discuss how and why the data is transformed from one representation to the next. Activity space contains experimental observations of neuronal activity. A vector in this space represents the instantaneous activation of all individual neurons at a single time point. Each component of this vector represents the instantaneous activity of a single identified neuron. The ultimate goal of the method is to efficiently model the temporal sequences of neuronal activation. To do this, we first need to extract relevant variables sufficient to fully describe the dynamics which give rise to neuronal activity. This collection of variables is known as the phase space. In phase space each dimension represents a unique relevant variable. In contrast to neuronal activity, these variables may not necessarily be directly observed. We will approximate the time evolution of the system in phase space by constructing a transition probability matrix. Each element $(i, j)$ of this matrix corresponds to the probability that a system observed at location $i$ in phase space will transition to location $j$ after one time step (see below, *Figure 4—figure supplement 2*). This $n \times n$ representation, where $n$ is the number of observations, gives an approximation of the velocity of the system at each observed point in phase space. Finally, we will simplify this table of velocities to extract manifold space – allowing for a minimal representation of the dynamics. Temporal evolution of the system in the manifold space can then be readily simulated to yield quantitative predictions about future neuronal activity.

### From activity space to phase space

The global dynamics of a nervous system depends on biophysical processes beyond neuronal firing. It is experimentally intractable to record all such processes including time and voltage dependent

currents, neurotransmitter and neuromodulator release, hormonal signaling, plasticity, etc. However, the key variables that make up phase space can be extracted from the observations using methods known as delay embedding (*Takens, 1981*; *Packard et al., 1980*). The main idea behind delay embedding is that one can use the experimental observations (neuronal activation and its time-deriv-ative) to extract independent measurements that together form the phase space. To extract inde-pendent measurements from a single time series (e.g. neuronal activity), the delay time τ is chosen such that correlation between two points in the activity space separated by τ is negligible. These delayed versions of the time series correspond to different dimensions of the reconstructed phase space. According to Takens' theorem (*Takens, 1981*), this reconstructed space preserves essential features of the dynamics which are required for model construction. When phase space is well approximated, points that are close to each other have similar velocities. Consequentially, if two tra-jectories in the time series data are close in phase space they will continue to evolve in time along similar trajectories – giving rise to recurrent coherent trajectories in the dynamics. The process of delay embedding dramatically inflates the dimensionality of the data making it unusable for complex time series such as activation of many neurons. Thus, the final critical step of the method will reduce the dimensionality of the system.

## From phase space to transition probability matrix

The goal of this step will be to enumerate the phase space dynamics into a discrete transition prob-ability matrix $\mathbf{M}$. The $i^{th}$ row of this matrix tabulates the probability that a system starting out in state $i$ will transition in any other state $j$ after one time step. In this case, the state of the system is described by delay embeddings of observed neuronal activity. To assign transition probabilities, we use diffusion mapping (*Nadler et al., 2006*; *Coifman and Lafon, 2006*; *Lian et al., 2015*) – a non-lin-ear dimensionality reduction technique. Similar to local linear embedding (*Roweis and Saul, 2000*) or isomap (*Tenenbaum et al., 2000*), diffusion maps seek to preserve local relationships between nearby points. Points that are close together in phase space will be assigned high transition proba-bilities. However, points that are far away (*Equation 13*) in phase space are not directly connected (i.e. transition probability is zero). After appropriate normalization which ensures that the sum of all probabilities in a row adds up to 1, this diffusion map can be used to simulate the time evolution of the system. To simulate evolution after $N$ time steps $\mathbf{M}$ is exponentiated $N$ times. In standard diffu-sion maps the transition probabilities between points are assumed to be symmetric (i.e. transition probability $P_{i \to j} = P_{j \to i}$). Yet, this approach does not take into account the fact that neuronal activity is ordered in time. We therefore modify the transition probability calculation to include temporal information. To take temporal information into account, we compute transition probability between the state of the system $\mathbf{D}_t$ at time $t$ and points in the neighborhood of the next experimentally observed state $\mathbf{D}_{t+1}$. These transition probabilities are computed as a Gaussian centered at $\mathbf{D}_{t+1}$,

$$k_{FP}(\mathbf{D}_t, \mathbf{D}_j) = exp\left(-\frac{\|\mathbf{D}_{t+1} - \mathbf{D}_j\|_2^2}{2\sigma^2}\right),$$

(1)

where $\mathbf{D}_j$ is a point in the local neighborhood of $\mathbf{D}_{t+1}$, and $\|\cdot\|_2^2$ is the Euclidean distance. $\sigma^2$ is a normal-ization term that sets the size of the local neighborhood (see below for details). The result is that time evolution of neuronal activity given by asymmetrical $\mathbf{M}$ preserves the temporal order of neuronal activity.

## From transition probability matrix to manifold space

Although $\mathbf{M}$ can be used to simulate neuronal activity, it is not in itself a particularly useful model. $\mathbf{M}$ does not directly inform dominant features of neuronal dynamics and simulations of $\mathbf{M}$ can only generate reordered versions of the experimentally observed time series. This limitation is due to the fact that $\mathbf{M}$ is only defined in terms of the observed states of the system. However, spectral analysis of $\mathbf{M}$ can be used to extract salient features of neuronal dynamics (fluxes). Because $\mathbf{M}$ is not sym-metrical, it can give rise to rotational dynamics. To identify the most salient rotational fluxes, we per-form spectral analysis of $\mathbf{M}$ (see below). As a result, each point in $\mathbf{M}$ is assigned a phase along the rotational flux. To identify the most dynamically salient fluxes, we find the complex eigenvalues of $\mathbf{M}$ with the largest modulus. A pair of complex conjugate eigenvectors associated with this eigenvalue

relate states of the nervous system $\mathbf{D}_t$ to the phase of the rotational flux. This allows us to bin points with similar phase. Because in *C. elegans* there are multiple rotational fluxes, it is not a priori clear which rotational flux is associated with a given phase. This can be resolved using clustering analysis of $\mathbf{M}$ (see below). As a result of eigendecomposition and clustering, each point in $\mathbf{M}$ is assigned to a single bin defined by the identity of the flux and the phase along it. We refer to the transition probability matrix simplified in this fashion as the manifold. Simulations of the manifold are sufficient to predict behavioral statistics, sequences of behaviors, timing of individual behavioral transitions, and neuronal activation. Furthermore, simulations in manifold space yield novel neuronal activity patterns not directly observed in the experiment.

## Origins of cyclic fluxes in neuronal dynamics

In this section, we will present a theoretical argument (see also) (*Wang et al., 2008*) which suggests that cyclic fluxes are likely to be a common feature of neuronal dynamics. This argument motivates the manifold reconstruction method (see below).

Neuronal systems are inherently noisy. Thus, the most sensible approach is to model the dynamics of the nervous system using both deterministic dynamics and stochastic processes (*Yan et al., 2013*),

$$\frac{d\mathbf{X}}{dt} = \mathbf{F}(\mathbf{X}) + \epsilon, \tag{2}$$

where $\mathbf{F}(\mathbf{X})$ is the driving force which quantifies the deterministic aspect of neuronal dynamics, $\mathbf{X}$ is the position in state space, and $\epsilon$ is noise. Because of noise, it is not possible to precisely model the trajectory of any single point starting out at some location in $\mathbf{X}$. It is possible, however, to model the temporal evolution of a cloud of points – or more precisely a probability distribution of points – $P(\mathbf{X})$ (*Pathria, 1996*). We begin with the law of probability conservation,

$$\frac{dP(\mathbf{X})}{dt} = -\nabla\mathbf{J}(\mathbf{X},t), \tag{3}$$

which states that the change in probability $P$ is due to the local flux, $\mathbf{J}(\mathbf{X},t)$, in that region. In systems with homogeneous (constant in space) noise, the flux is defined by:

$$\mathbf{J}(\mathbf{X},t) = \mathbf{F}(\mathbf{X})P - \nabla P, \tag{4}$$

where $\mathbf{F}(\mathbf{X})$ is the driving force. We now assume that the system is at steady state during the time course of the experiment. Mathematically, this corresponds to the assumption that probability distribution is constant. Thus,

$$\frac{dP(\mathbf{X})}{dt} = 0 = \nabla\mathbf{J}(\mathbf{X},t). \tag{5}$$

From a neuroscience standpoint, this statement corresponds to the assumption that the nervous system is not changing (e.g. learning) during the experiment. This is a reasonable assumption for the datasets in this manuscript which last ~15 min per recording. Over long-term recordings, this assumption can be invoked in a piecewise fashion over shorter time intervals.

One well-known solution to *Equation 5* is a purely stochastic case where the deterministic flux of the system vanishes at all $\mathbf{X}$, $\mathbf{J}(\mathbf{X},t) = 0$. In this case, the only meaningful measure of neuronal activity is the probability of different activity patterns. This assumption is invoked in stochastic models of neuronal activity such as maximum entropy models (*Tkačik et al., 2013*; *Tang et al., 2008*), Hopfield networks (*Hopfield, 1982*), and others.

Yet, another class of solutions exist when the flux does not vanish at steady state (*Yan et al., 2013*). The key insight is that in order to keep the distribution of states $P(\mathbf{X})$ constant, the flux must be purely cyclic,

$$\mathbf{J}(\mathbf{X},t) = \nabla \times A, \tag{6}$$

where $A$ is an arbitrary vector field. Such fluxes are divergence free, and will always form complete loops. Intuitively, this means that a system that evolves around a cyclical orbit will at once have a

deterministic flux $J(\mathbf{X}, t) \neq 0$ and satisfy the steady state assumption. For such systems, the driving force is

$$\mathbf{F}(\mathbf{X}) = \mathbf{D}\frac{\nabla P(\mathbf{X})}{P(\mathbf{X})} + \frac{\mathbf{J}(\mathbf{X})}{P(\mathbf{X})}, \tag{7}$$

where $\mathbf{J}(\mathbf{X})$ is the flux at steady state. Note that *Equation 7* is a form of the Fokker-Planck equation. The driving force is made of two distinct terms. The first term corresponds to diffusion, while the second corresponds to a deterministic cyclic flux. The purpose of the manifold reconstruction method is to discover this deterministic cyclic flux in neuronal recordings.

## Extracting cyclic flux from data

The ultimate goal of the manifold extraction method is to express neuronal dynamics as a linear stochastic dynamical system. This requires the construction of a transition probability matrix $\mathbf{M}$ based on empirical observations of neuronal activity where each element $(i, j)$ is given by,

$$\mathbf{M}_{ij} = \frac{\|s_i \rightarrow s_j\|}{\|s_i\|}, \tag{8}$$

where $\|s_i \rightarrow s_j\|$ is the number of times the system transitions from state $i$ to state $j$ and $\|s_i\|$ is the total number of times the system is found in state $i$ can be used to simulate the time evolution of the system for a single time step by

$$\mathbf{X}_{t+1} = \mathbf{M}\mathbf{X}_t, \tag{9}$$

or for some arbitrary time $t$

$$\mathbf{X}_t = \mathbf{M}^t\mathbf{X}_o, \tag{10}$$

where $\mathbf{X}_o$ is the initial state of the system. Alternatively this equation can be rewritten in terms of the eigenmodes of $\mathbf{M}$,

$$\mathbf{X}_t = \sum_i c_i \lambda_i^t \phi_i, \tag{11}$$

where $\lambda_i$ are the eigenvalues, $\phi_i$ are the eigenvectors and $c_i$ is the projection of the initial state of the system onto the $i$-th eigenvector. Under a broad range of conditions, the largest eigenvalue of $\mathbf{M}$ is $\lambda = 1$. This corresponds to an assertion that such systems come to a single steady state. The associated eigenvector corresponds to the steady state distribution of the system. If $\mathbf{M}$ is symmetrical, that is $\mathbf{M}_{i,j} = \mathbf{M}_{j,i}$, then all eigenvalues of $\mathbf{M}$ are real, and the resulting system is purely stochastic. Asymmetry can give rise to complex eigenmodes. Then *Equation 11* becomes an equation of a decaying wave in the plane spanned by a pair of complex conjugate eigenvectors. These decaying spirals correspond to the cyclic fluxes of *Equation 7*. In the long time limit, all eigenmodes with complex eigenvalues whose modulus is much less than one damp out. Complex modes with eigenvalues near one heavily shape the dynamics of the system even in the long time limit. These eigenmodes are used to identify the cyclic fluxes of neuronal activity.

In order to construct $\mathbf{M}$, two steps are required: definition of the state of the system and definition of distances between two points in the state space. The distances between points in state space are used to define transition probabilities. We extract state space from the data using delay embedding (see below), and then use diffusion mapping to define distances between points in the delay embedded coordinates.

## Delay embedding to uncover true phase space

There are several algorithms for finding a good delay embedding parameters and number of delays (*Packard et al., 1980*; *Sauer et al., 1991*; *Buzug and Pfister, 1992*). The key point is that maximally independent measurements are chosen. Here, we used autocorrelation as a measure of interdependence to estimate delay $\tau$ such that autocorrelation becomes ~0. For *C. elegans* manifolds, we used $\tau = 10$ frames (~4 s.). We explored a range of number of delays. The number of delays used to generate the figures is five but the results are fairly robust to changes in this parameter. *Kato et al.*

*(2015)* notice that derivatives of neuronal activity in *C. elegans* are useful for analysis of neuronal dynamics. Building upon their result here, we used the adjoint space formed by the raw neuronal activity and its derivative (akin to position and velocity). At every time $t$, the position of the system in the raw neuronal activity space $\mathbf{A}_t$ can be mapped to the delay embedded space $\mathbf{D}_t$ using the following formula:

$$\langle \mathbf{A_t} \ldots \mathbf{A_{t-5\tau}}, \mathbf{A'_t} \ldots \mathbf{A'_{t-5\tau}} \rangle \rightarrow \mathbf{D_t}, \tag{12}$$

where $\mathbf{A_t}$ is a snapshot of neuronal activity, $\mathbf{A'_t}$ is a snapshot of the derivative of neuronal activity, $\langle \ldots \rangle$ denotes concatenation of vectors, and $\mathbf{D_t}$ is the position of the system in the delay embedded coordinates at time $t$.

## Diffusion mapping

As discussed in the manuscript, delay embedded neuronal activity of even simple nervous system of *C. elegans* is too high dimensional to be useful for characterizing system dynamics. For instance for the common 15 neuron dataset $\mathbf{D}_t$ is a 180-dimensional vector. Yet, as has been shown by *Coifman and Lafon (2006)*, there is a fundamental connection between the eigenvectors of the Markov chain (*Equation 11*) and dimensionality reduction. This connection is the motivation for a class of methods known as diffusion mapping. The basic idea behind diffusion map is to cast distances between two nearby points in state space as transition probabilities (*Nadler et al., 2006*; *Coifman and Lafon, 2006*). Diffusion maps have two fundamental advantages: they are nonlinear and preserve local structures. The former is critical here because neuronal dynamics can be safely assumed to be nonlinear. The latter is important because large distances in complex high-dimensional and nonlinear datasets are meaningless (*Aggarwal et al., 2001*). This local geometry assumption is common to a number of nonlinear dimensionality reduction techniques such as isomap, locally linear embedding, and kernel PCA. Traditional applications of diffusion maps have been in dimensionality reduction. For these purposes, the diffusion map is assumed to be symmetric. Here, we modify the formalism slightly to account for the possibility of cyclic fluxes and therefore allow for the possibility of asymmetry in the transition probabilities $i \rightarrow j$ and $j \rightarrow i$.

This asymmetry arises naturally if the diffusion map is constructed such that experimentally observed order of neuronal activation is preserved. We accomplish this simply by centering the kernel of a diffusion map $k_{FP}$ on the next empirically observed data point as follows:

$$k_{FP}(\mathbf{D}_t, \mathbf{D}_j) = exp\left(-\frac{\left\| \mathbf{D}_{t+1} - \mathbf{D}_j \right\|_2^2}{2\sigma^2}\right), \tag{13}$$

where $\mathbf{D}_t$ is the position of the system in the delay embedded coordinates at time $t$, $\mathbf{D}_{t+1}$ is the next empirically observed state of the system, $\mathbf{D}_j$ is a point in the local neighborhood of $\mathbf{D}_{t+1}$, and $\|\cdot\|_2^2$ is the Euclidean distance. $\sigma^2$ is a normalization term that sets the size of the local neighborhood. The key mathematical insight is that after appropriate normalization, diffusion maps converge to the Fokker-Planck (*Nadler et al., 2006*) operator. Under these conditions, *Equation 9* is an approximation of *Equation 7* and thus diffusion maps constitutes a natural way to cast distances between points along a trajectory generated by a stochastic dynamical system. To see this, note that if the local neighborhood is decreased such that it only contains a single point $\mathbf{D}_{t+1}$, *Equation 13* will exactly reproduce the observed neuronal activity in the correct temporal order. In other words the matrix $\mathbf{M}$ constructed by applying *Equation 13* to all pairs of states will have 1's for all $\mathbf{M}_{i,i+1}$ and zeros elsewhere.

This matrix, however, is not particularly useful for simulating neuronal dynamics because it will only exactly recapitulate experimental observations. To overcome this limitation, normalization term, $\sigma^2$, sets the amount of noise around the experimentally observed neuronal trajectories and allows the simulation to deviate from the actual experimental measurements. Although it is likely that several choices of $\sigma^2$ will work, here we chose

$$\sigma^2 = \sigma_l(\mathbf{D}_{t+1})\sigma_l(\mathbf{D}_t)\langle k_{FP}\rangle_{\mathbf{XY}}, \tag{14}$$

where $\sigma_l(\cdot)$ is the standard deviation of the data in a 12 time-step temporal window centered at time

$t$, and $\langle k_{FP} \rangle_{\mathbf{XY}}$ is the mean value of the kernel (*Equation 13*) over all data points in the neighborhood of $\mathbf{D}_{t+1}$.

For *C. elegans*, we compute $k_{FP}$ for the 12 nearest neighbors to each point $\mathbf{D}_{t+1}$. The method is robust to the exact number of nearest neighbors used.

*Equation 13* was then evaluated for all observed states of *C. elegans* neuronal activity. This results in an $n \times n$ (where $n$ is the number of delay embedded snapshots of neuronal activity) matrix. This matrix is normalized such that the sum along each row is equal to 1. This normalization converts the distance matrix to a right stochastic (Markov) matrix $\mathbf{M}$. The complex eigenvalue with the largest modulus of $\mathbf{M}$ defines the dominant cyclic flux. The projection of the associated pair of complex conjugate eigenvectors onto elements of $\mathbf{M}$ define the phase along the cyclic flux $\theta$ associated with each delay embedded neuronal activity state.

## Trajectory clustering

If there are multiple cyclic fluxes as in *C. elegans* CNS, then in addition to the phase one needs to also know the identify of the flux. To identify fluxes we preform clustering on the data. Any standard clustering algorithm will suffice, and this section will only detail one of many possible choices (*Rubinov and Sporns, 2010*) that can be used. We did not explore the effects of the choice of clustering and suspect that, as is the case with many clustering applications, the best choice will depend on the specifics of the dataset. We use a maximum modularity algorithm (*Newman, 2006*) on the transition probability matrix constructed according to *Equation 13*. By construction, the transition probability matrix is sparse (only transitions in local neighborhoods are allowed). Therefore, in its raw form the system given by this matrix will not explore the manifold sufficiently as it will be trapped in each individual isolated neighborhood. To overcome this problem, the matrix is exponentiated N times until a minimum fraction of elements of each row are non-zero (25% in the *C. elegans* data). Conceptually, this corresponds to finding the evolution of the system after $N$ time steps and is closely related to the 'diffusion distance' (*Coifman and Lafon, 2006*). Specific choice of $N$ does not have a strong influence on the results, so long as the resultant matrix is not too sparse. Note that the exponentiation of $\mathbf{M}$ does not change its eigenvalues.

Two major features are found in the transition probability matrix (*Figure 4—figure supplement 2*): patches and diagonals. Square patches identify locations where the system exhibits Brownian motion near a point attractor. In these patches, the matrix is approximately symmetric and therefore stochastic processes dominate. Diagonal traces identify coherent trajectories where deterministic fluxes are dominant.

The square patches are already suitable for clustering. If two elements of the matrix belong to the same point attractor, they will be found in the same square patch. The situation is slightly more complex for coherent trajectories identified by diagonal bands. To determine whether two elements of state space belong to the same coherent trajectory, we compute the maximum correlation of each row (distances from each element of state space) and time lagged copies all the other rows $max(corr(row_i, shift(row_j, t)))$. Where $row_i$ is the $i$th row, shift moves all elements in the row $t$ steps to the right and the maximum is taken over all $t$. This newly formed matrix has the same dimensions as the original transition probability matrix. We apply standard maximum modularity clustering using the *community_louvain* function from the Brain Connectivity Toolbox to this matrix (*Rubinov and Sporns, 2010*). Clustering assigns flux ID $\alpha$ to each point $\mathbf{D}_t$ in the delay embedded neuronal activity space. Together with the phase $\theta$, assigned by eigenmode decomposition, $(\theta, \alpha)$ span the phase space of neuronal dynamics.

## Manifold reconstruction

The phase space spanned by $\theta$ and $\alpha$, rather than raw neuronal activity provide a proper basis with respect to which neuronal activity can be averaged. These averages are shown as manifolds in *Figure 4*. For each $\alpha$, we sort the delay embedded neuronal activity according to its phase $\theta$. We then convolve this activity with a sliding Gaussian window over $\theta$. The width of the Gaussian smooths neuronal activity but does not play any appreciable role in setting the dynamics over a broad range of values. To visualize these $\theta$-averaged trajectories in *Figure 4*, we project them onto the first three principal components.

Because phase identity $\alpha$ is discrete, $\theta$-averaged trajectories form disjoint bundles. For the purposes of visualization (*Figure 4*) these bundles are joined together by interpolating a spline (over both position and direction) from the end of one bundle to the beginning of the next bundle. This interpolation is performed solely for visualization and plays no role in quantitative analyses – which are all done in the manifold space $(\theta, \alpha)$.

## Two neuron toy system

We make use of a network of two neurons (*Appendix 1—figure 1*) whose simplified biophysics are modeled by *Ermentrout (1998)*; *Beer (1995)*

$$\frac{dA}{dt} = -A + \sigma(8A - 6B - 0.34) + \epsilon, \tag{15}$$

$$6\frac{dB}{dt} = -B + \sigma(16A - 2B - 2.5) + \epsilon, \tag{16}$$

$$\sigma(x) \equiv 1/(1 + \exp(-x)), \tag{17}$$

where the noise term, $\epsilon$, is drawn independently from a Gaussian distribution $\epsilon \sim \mathcal{N}(0, 0.1)$ at each time step. A schematic of the system, along with an illustration of the asymptotic behavior of the dynamics are given in *Appendix 1—figure 1—figure supplement 1*. *Appendix 1—figure 1—figure supplement 1D* shows an example trace used in the construction of the manifold in *Appendix 1—figure 1*.

## Calcium imaging

Here, we analyze $Ca2^+$ imaging data published by *Kato et al. (2015)* and *Nichols et al. (2017)*. The deviation of fluorescence from baseline ($\Delta F/F$) is considered as a proxy for neuronal activity. The manifold was constructed on the data from *Kato et al. (2015)*. The validation of the predictions concerning timing of behavioral switching was performed using the dataset from *Nichols et al. (2017)*. The dataset were obtained as MATLAB files and were preprocessed by the Zimmer Lab to account for the effects of bleaching.

*C. elegans* were immobilized in a microfluidic device (*Schrödel et al., 2013*) under environmentally constant conditions. The 107 to 131 neurons detected in each worm in the *Kato et al. (2015)* span all head ganglia, all head motor neurons and most of the sensory neurons and interneurons along with most of the anterior ventral cord motor neurons (*White et al., 1986*). Of the identified neurons for each worm there is a subset of 15 neurons (AIBL, AIBR, ALA, AVAL, AVAR, AVBL, AVER, RID, RIML, RIMR, RMED, RMEL, RMER, VB01, VB02) which were unambiguously identified in each worm. This set of neurons is used to build the manifold.

We adopt the same behavior states defined by *Kato et al. (2015)*. The four primary behavioral states are forward locomotion, turns (FALL), reversals (RISE) and backwards locomotion (*Figure 1*). FALL and RISE were further split into two distinct motor command states by performing k-means clustering on the RISE and FALL phase timing vectors separately. More details of the experiment can be found in *Kato et al. (2015)*. All analyses were implemented in MATLAB. In addition to the processing steps by *Kato et al. (2015)* which account for bleaching of the GCaMP proteins, we smooth the ($\Delta F/F$) time series for each neuron with a Gaussian filter ($\sigma = 1$) and convert the filtered time series to z-scores. Note that the amount of smoothing applied is orders of magnitude less than the autocorrelations found in the data (*Figure 2—figure supplement 2*). Because the experimental data are dominated by forward and backward locomotion, we focus our predictions on just these two behaviors. We do not attempt to predict dorsal or ventral turns or the two types of reversals (1 and 2) defined by Kato et al. because these behaviors occupy a small fraction of the observed time series.

## Analysis of neuronal activity trajectories

To compare neuronal activity in different instances of the same type of locomotor behavior, we convert from raw time to 'behavioral phase' $\phi_b$ as follows $\phi_b = (t_i - t_{start})/(t_{end} - t_{start})$ where $t_i$ is the raw time, $t_{start}$ and $t_{end}$ are the beginning and end times of the behavior respectively. This time warping

normalizes $\phi_b$ such that it ranges from 0 to 1 (beginning and end) of each individual instance of behavior. In order to average neuronal activity across different instances of the same behavior, we sample $\phi_b$ in equally spaced 100 intervals. Prior to averaging neuronal activity, constant shift in the $\Delta F/F$ signal was subtracted (i.e. the mean of neuronal activity across $\phi_b$ for each individual instance of behavior is zero). Thus, differences in neuronal activity between two different individuals reflect differences in the temporal pattern of activation rather than shifts in the overall level of activity. Neuronal activity normalized in this fashion and averaged across instances of a particular locomotor behavior in each animal is plotted as a function of $\phi_b$ in *Figure 1*.

Lack of overlap between 95% confidence intervals around the mean neuronal activity observed in different animals in *Figure 1* signifies statistically significant differences between neuronal activity in different individual *C. elegans*. To quantify these differences for each neuron and each type of locomotion, we constructed an $n \times m$ matrix $T$, where $n$ is the number of instances of behavior (observed in all 5 *C. elegans*) and $m$ is the number of $\phi_b$ bins. Because neuronal activity is smooth, activation in nearby phases is highly correlated. To remove these correlations, $T$ was subjected to principal component analysis (PCA) and projected onto the first principal component (PC1). This results in $n$ scalars (one for each neuronal activity trajectory). This quantity reflects the similarity between projections onto the first principal component (mean neuronal activity trajectory across all animals shown by dashed line in *Figure 1*) and each individual cycle of behavior. We subjected this PC1 projection to a one-way ANOVA (with animal ID as the categorical variable). p-values for ANOVA obtained for each combination of locomotor behavior and neuron ID are shown in *Figure 1*. For statements concerning statistical significance in the text we used ($\alpha$=0.05) after a Bonferroni correction for multiple comparisons.

## Decoding behavior on the basis of neuronal activity

A subset of neuronal activity was chosen as the training set while the remaining neuronal activity were used as a validation set. A template was constructed by averaging activity of each neuron at the onset of each backing behavior in the training set. This template was convolved with neuronal activity in the validation set to yield similarity score between the template and neuronal activity at each time point in the validation dataset. For the decoding in *Figure 1*, we chose a threshold of this score such that the overlap between distribution of scores associated with true events (initiation of backing behavior) and distribution of scores of false events (all other behaviors) is minimized (see below). To minimize the effect of noise and compensate for the low probability of true positives, we only considered local maxima of the score. To compensate for the inherent imprecision of assigning behavioral states we considered all peaks found within 10 frames ($\approx 3s.$) of the initiation of backing behavior as true events.

The probability of correctly identifying a behavioral event given a specific threshold $X_{thres}$ is

$$p(\theta = 1 \mid X \geq X_{thres}) = \frac{\|X_{\theta=1} \geq X_{thres}\|}{\|X_{\theta=1}\| + \|X_{\theta=0} \geq X_{thres}\|}, \tag{18}$$

where $\theta = 1$ are true events, $X_{\theta=1}$ are the scores of true events, $X_{\theta=0}$ are scores of false events, and $\|\cdot\|$ denotes the number of elements in the set. Optimal threshold $\widetilde{X}_{thresh}$ is found as argmax of *Equation 18* with respect to $X_{thresh}$ in each training dataset individually and used to compute correct decoding probability in *Figure 1*. For single animal predictions, 1/2 of the backing behaviors in each animal was used as the training dataset while the remaining 1/2 of backing behaviors in the same animal was used as validation. For the cross animal predictions, we used 1/2 of the backing behaviors in four animals to construct a training set and used the 1/2 of the events in the left out animal as the validation dataset. For the shuffled control, we used random time points as true events in the training dataset. To obtain errors around decoding probability in *Figure 1*, we bootstrapped this procedure for multiple partitions of the data into training and validation datasets. Box plot in *Figure 1* shows the distribution of the decoding probability across all bootstraps.

## Manifold behavioral statistics

Manifold space was divided into Gaussian bins each centered at a particular phase $\theta$ where $\Delta \theta \approx 0.05$. Total of 426 bins were used for the entire data set. The likelihood that a given point $\mathbf{D}_t$ belongs to each $\theta$ bin was computed and $\mathbf{D}_t$ was assigned to the most likely bin. Each point $\mathbf{D}_t$ was assigned a

behavioral state by *Kato et al. (2015)*. Thus, for each $\theta$ bin, we attain a distribution of assigned behaviors. This distribution is encoded in color of the manifold (*Figure 4*). If $\theta$ did not reflect behaviorally relevant information, then the distribution of behavioral states in $\theta$ will be the same as in the dataset as a whole. This constitutes the null hypothesis against which manifold-based decoding of behavioral state were tested.

Similar approach was taken for the 'leave one out' manifold prediction. Manifold was constructed as above on the basis of data from four animals from Kato et al. Distribution of behavioral states for each $\theta$-bin was estimated on the basis of only these four animals. Then data from the fifth animal left out of manifold construction was delay embedded as above. Each snapshot of delay embedded activity of the fifth worm was assigned to the nearest $\theta$-bin. In an attempt to decode the behavioral state of the fifth worm, in each $\theta$-bin we compare the behavioral state of the left out animal to the most likely behavioral state in the $\theta$-bin comprised of data from the remaining four animals. The null hypothesis relative to which quality of decoding was compared is that the prevalence of each behavioral state in a given $\theta$-bin is the same as the prevalence of the behavioral state in the whole dataset. This procedure was repeated by leaving out the data from each one of the five worms in the Kato et al. dataset in turn. The distributions of behavioral states from left out animals and the other four animals used to construct the manifold was compared using $\chi^2$. $\chi^2$ averaged over all $\theta$-bins and all five left out animals is reported in the manuscript.

## Manifold behavioral dynamics

As a result of the manifold construction method (see above), each point in the observed neuronal time series $\mathbf{D}_t$ is assigned to a single bin in the manifold space $(\theta_i, \alpha_i)$. Thus, rather than describing the time series in terms of activation of neurons, we have a 2-D description of the state of the system at each point in time. This allows us to directly estimate transition probability between two states $(\theta_i, \alpha_i) \rightarrow (\theta_j, \alpha_j)$ by *Equation 8*. The time evolution of the system can now be readily simulated using *Equation 9*. This simulation gives rise to a new time series. To map from manifold space back to neuronal activity or behavior, each point in $(\theta, \alpha)$ is assigned to a (delay embedded) neuronal activity by reversing the relationship in *Equation 12*. Recall that each point in manifold space $(\theta, \alpha)$ corresponds to a cloud of points in the delay embedded space. Here, for the purposes of simulation of neuronal activity we parsimoniously assigned each point in $(\theta, \alpha)$ the mean of the delay embedded neuronal activity that was assigned to this bin. Alternatively, a random sample from this distribution of points can be chosen. Behavioral state that is associated with this *newly* simulated neuronal activity snapshot was assigned by sampling the distribution of behavioral states in each phase bin.

## Dwell time statistics

Simulated dwell time statistics are calculated by assigning to each time point in the simulation a behavior based on the most prevalent behavior in that time point's corresponding phase bin. This behavioral time sequence is then smoothed by a median filter with a size of 11 time steps (~3.5 s.). Turns and reverses are transients and constitute a small fraction of the dataset and are thus highly under sampled. Thus, we restrict our analysis to only forward and backwards locomotion. Backing bouts are periods in which the animal sustains backing locomotion with minimal forward locomotion. These events are defined as periods in which the forward locomotion state fails to last for more than 30 frames (~10 s). Dwell time distributions are shown in *Figure 2*.

Experimentally observed and simulated dwell time histograms are smoothed using the ksdensity function in MATLAB. $r^2$ values are calculated from these smoothed histograms.

## Time to transition analysis

Each data point $\mathbf{D}_t$ is characterized by two independent quantities: time since the onset of the behavior $t_{start}$ and position in manifold space $(\theta_i, \alpha_i)$. The null hypothesis is the expected time to behavioral transition is based solely the dwell time distribution. This corresponds to finding the survival function given by the right tail of the dwell time distribution from $t_{start}$ to infinity,

$$P_{null}(t) = P(t + t_{start}), \tag{19}$$

where $P(t)$ is the probability of the transition occurring at time $t$ is calculated by averaging the time since the onset of the behavior over all points in a given phase bin $(\theta_i, \alpha_i)$. To find the corresponding

manifold-based prediction the distribution of times until behavioral switch is explicitly found in the simulated neuronal activity data. We identify all points belonging to a particular phase bin and determine the distribution of times until the behavior is terminated.

In order to apply the same analysis to the data presented in *Nichols et al. (2017)*, we restricted our analysis to the prelethargus N2 animals (n = 11). These are most genetically similar to the five animals used in the construction of our manifold. First, we selected the subset of neurons that were uniquely identified in each animal from the *Nichols et al. (2017)* dataset and the 15 neurons on the basis of which the manifold was constructed using the Kato et al. data. The number of common neurons varied between 8 and 13. Neuronal activity from Nichols et al. was delay embedded as above yielding a set of $\mathbf{D}_t$s. Each $\mathbf{D}_t$ from the validation dataset was assigned to the closest phase bin as in the 'leave one out' validation. The only exception here is that the distance to the closest phase bin was computed by omitting the neurons that were missing from the animal in the validation dataset. The distribution of times to behavioral transition in the validation dataset was empirically estimated by observing the switching times of all points in the validation dataset assigned to a given phase bin $(\theta_i, \alpha_i)$.

The *Nichols et al. (2017)* animals use a different behavioral assignment paradigm than those in *Kato et al. (2015)* and so we normalized the behavioral assignments by assigning forward locomotion to any point in which the z-score of AVAL was below a given threshold and backward locomotion to any point in which the z-score of AVAL was above that threshold. This method does not preserve finer details such as the timing of turns and reversals, and so our predictions do not attempt to address those behaviors.

## Relative information calculation

Starting from the motor command dwell time distributions as shown in *Figure 2C* we calculate the Kullback–Leibler divergence between the experimentally observed distribution and the simulated distribution. Because the exact binning heavily effects information theoretic quantities such as KL divergence, we scan over a range of bin counts between 40 and 200 and choose the minimum KL divergence in this range. Finally, to normalize these quantities for easy comparison we calculate the ratio between the original total information and the modified model (different numbers of neurons or different parameters) by:

$$I_{tot} = \sum_{b \in \mathcal{B}} D_{KL}(P_{obs} || P_{sim}), \tag{20}$$

$$I_{rel} = I_{tot}^{original} / T_{tot}^{modified}, \tag{21}$$

where $D_{KL}$ is the KL divergence, $\mathcal{B}$ is the set of three motor command distributions in *Figure 2C* (forward locomotion, backwards locomotion and backwards bouts) and $P_{obs}$ and $P_{sim}$ are the observed and simulated dwell time distributions respectively.

## Predictions without AVA

For the robustness tests in *Figure 2—figure supplement 4* and *Figure 3—figure supplement 1*, the models are built with data from AVA excluded. For the calculation of motor command dwell time distributions, the behaviors are assigned using the same behavioral assignment given by *Kato et al. (2015)*. For the predictions of behavioral switches presented in *Figure 3—figure supplement 1*, the mapping from neuronal activity space to manifold space does not make use of the data from AVA. However, we can include AVA in the mapping from manifold space back to neuronal activity space to recover the expected activity of AVA even though AVA was not explicitly used at any point in the model construction.

## Acknowledgements

We thank Sarah Friedensen, Adeeti Aggarwal, Guillermo Cecchi, Marcelo Magnasco, Drew Hudson, Tom Joseph, Manuel Zimmer, and Max Kelz for critically reading the manuscript. We also thank Manuel Zimmer and his lab for sharing their recordings of neuronal activity.

# Additional information

## Funding

| Funder | Grant reference number | Author |
|---|---|---|
| National Institute of General Medical Sciences | 1R01GM124023 | Alexander Proekt |

The funders had no role in study design, data collection and interpretation, or the decision to submit the work for publication.

## Author contributions

Connor Brennan, Conceptualization, Formal analysis, Visualization, Methodology, Writing—original draft, Writing—review and editing; Alexander Proekt, Conceptualization, Formal analysis, Supervision, Funding acquisition, Methodology, Writing—original draft

## Author ORCIDs

Connor Brennan (iD) https://orcid.org/0000-0002-5329-7720
Alexander Proekt (iD) https://orcid.org/0000-0002-9272-5337

## Decision letter and Author response

Decision letter https://doi.org/10.7554/eLife.46814.027
Author response https://doi.org/10.7554/eLife.46814.028

# Additional files

## Supplementary files

• Transparent reporting form
DOI: https://doi.org/10.7554/eLife.46814.017

## Data availability

Analysis code is available on github (URL: https://github.com/sharsnik2/AsymmetricDiffusionMapping; copy archived at https://github.com/elifesciences-publications/AsymmetricDiffusionMapping). All data used to construct the model was obtained from Manuel Zimmer's lab. The calcium imaging data used for modeling were described in Kato et al, Cell, 2015 and Nichols et al, Science 2017 and have been made available at https://osf.io/2395t/ and https://osf.io/kbf38/ respectively. From Kato et al. we used all datasets in the WT_NoStim.mat file (TS20141221b_THK178_lite-1_punc-31_NLS3_6eggs_1mMTet_basal_1080s, TS20140926d_lite-1_punc-31_NLS3_RIV_2eggs_1mMTet_basal_1080s, TS20140905c_lite-1_punc-31_NLS3_AVHJ_0eggs_1mMTet_basal_1080s, TS20140715f_lite-1_punc-31_NLS3_3eggs_56um_1mMTet_basal_1080s, TS20140715e_lite-1_punc-31_NLS3_2eggs_56um_1mMTet_basal_1080s). From Nichols et al. used all datasets in the n2_prelet.mat file (AN20150508a_ZIM504_46um_1mMTF_O2_s_21_PreLet_1530_, AN20150508b_ZIM504_46um_1mMTF_O2_s_21_PreLet_1530_, AN20150508j_ZIM504_46um_1mMTF_O2_s_21_PreLet_1910_, AN20150902b_ZIM945_1mMTF_O2_21_s_47um_1550_PreLet_, AN20150902d_ZIM945_1mMTF_O2_21_s_47um_1550_PreLet_, AN20150902e_ZIM945_1mMTF_O2_21_s_47um_1810_PreLet_, AN20150902h_ZIM945_1mMTF_O2_21_s_47um_1940_PreLet_, AN20150902i_ZIM945_1mMTF_O2_21_s_47um_1940_PreLet_, AN20160128c_ZIM1048_ilN2ce_NGM1mMTF_2010_PreLet_, AN20160128e_ZIM1048_ilN2ce_NGM1mMTF_2010_PreLet_, AN20160128f_ZIM1048_ilN2ce_NGM1mMTF_2115_PreLet_).

The following previously published datasets were used:

| Author(s) | Year | Dataset title | Dataset URL | Database and Identifier |
|---|---|---|---|---|
| Saul Kato, Harris S. Kaplan, Tina Schrödel, Susanne Skora, Theodore H. Lind- | 2015 | Whole brain imaging data from Kato et al | https://osf.io/2395t/ | Open Science Framework, 2395t |

| say, Eviatar Yemini, Shawn Lockery, Manuel Zimmer | | | | |
|---|---|---|---|---|
| Annika LA Nichols, Tomáš Eichler, Richard Latham, Manuel Zimmer | 2017 | Whole brain imaging data from Nichols et al. | https://osf.io/kbf38/ | Open Science Framework, kbf38 |

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

## Appendix 1

DOI: https://doi.org/10.7554/eLife.46814.018

### Extraction of the manifold from neuronal activity

We illustrate how neuronal dynamics can be extracted from neuronal activity using a simple system that consists of two reciprocally connected neurons $A$ and $B$ shown in *Appendix 1—figure 1—figure supplement 1*. The dynamics of the system exhibit two distinct oscillatory cycles *Appendix 1—figure 1—figure supplement 1* (*Ermentrout, 1998*; *Beer, 2000*) The first step in model construction is to find the essential variables that span the space where the dynamics unfold. For the model system, both $A$ and $B$ are required. However, even in the simple nervous system of *C. elegans* we cannot be certain that all of the relevant variables are observed (indeed only $\sim 1/20th$ of all neurons were reliably identified) or how they relate to experimental observations such as calcium signals measured using GCAMP. To mimic these conditions, we assume that only $A$ is experimentally observed. Fortunately, the information contained in $A$ can be used to reconstruct all of the relevant variables of the system using a class of methods called delay embedding (*Takens, 1981*; *Packard et al., 1980*; *Kantz and Schreiber, 2004*). To illustrate the geometrical intuition behind this method, note that when delayed values of $A$ are plotted against each other (*Appendix 1—figure 1B*), the trajectory traced by the system faithfully reconstructs two distinct cycles similar to those observed in the original system (*Appendix 1—figure 1—figure supplement 1*).

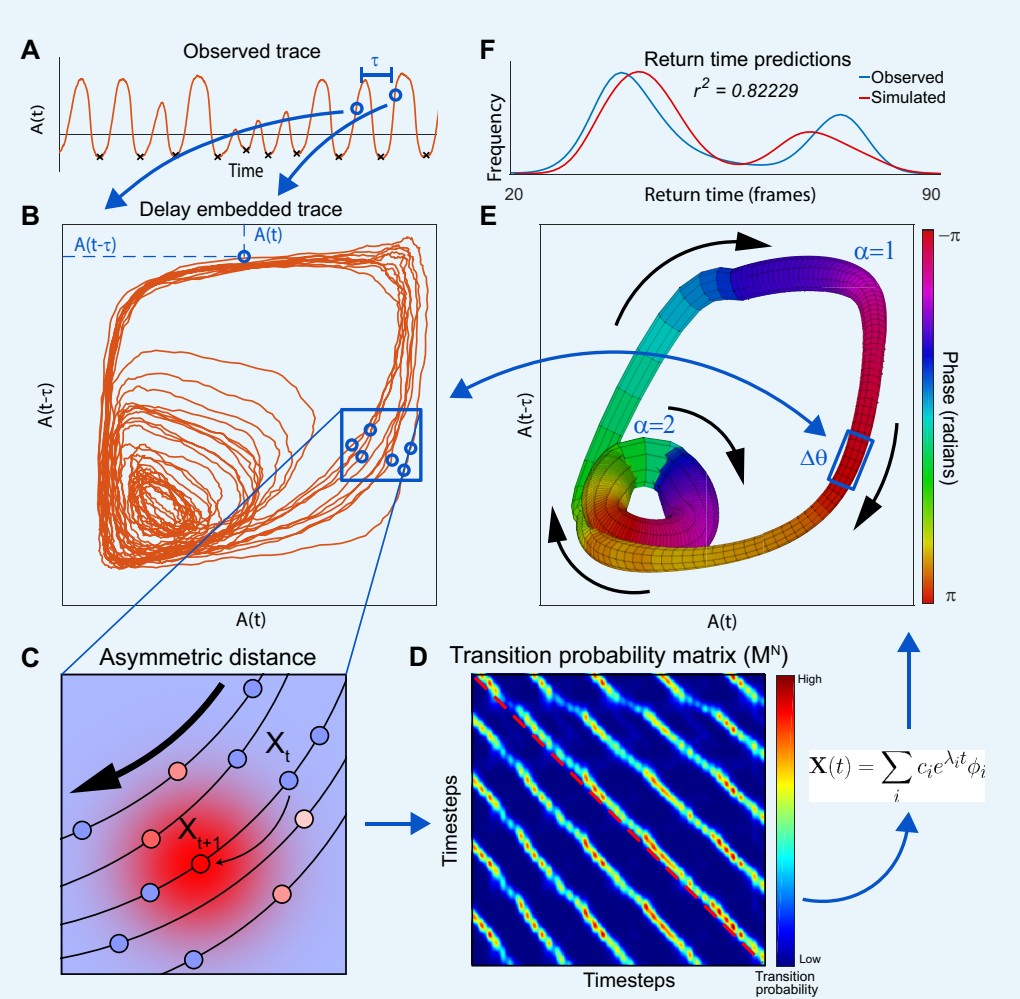

**Appendix 1—figure 1.** Extraction of manifold from incomplete neuronal activity data. (**A**) Observation of a single neuron A) from a two neuron system which produces two distinct modes of oscillation. Activity of $A$ at two time points separated by $\tau$, $A(t)$ and $A(t-\tau)$ (blue dots), is used as x and y coordinates of a single point in the reconstructed phase plane (blue arrows) obtained by delay embedding. (**B**) When the activity time series in A is projected onto the plane spanned by $A(t)$ and $A(t-\tau)$, the reconstructed system trajectory reveals two distinct oscillatory cycles present in the complete system. (**C**) Schematic of asymmetric diffusion map construction. The arrow indicates the direction of preferred flux. Transition probabilities from the state of the system at time $t$, $\mathbf{X}_t$, are computed using a local Gaussian kernel (red → high probability; blue → low probability). Because this kernel is centered on the next experimentally observed state $\mathbf{X}_{t+1}$, transition to $\mathbf{X}_{t+1}$ is most likely. However, transitions to parallel trajectories (red shows high probability of transition) are also possible. (**D**) Asymmetrical diffusion map $M$ computed for all pairs of states. $M$ is sparse because for each point, transition probability is nonzero just for $k = 12$ nearest neighbors. Exponentiation of $M$, which shows the evolution of the system after $N$ time steps, highlights the asymmetry. The red dashed line shows the main diagonal illustrating that the parallel bands are off center. Eigen-decomposition of the matrix reveals a linear sum of decay equations. $\lambda_i$ is the $i$-th eigenvalue and $\phi_i$ is the $i$-th eigenvector. $c_i$ represent the initial conditions of the system. (**E**) Manifold constructed for the two neuron system using only neuron A. Manifold construction method (Materials and methods) uses $\phi$ with the slowest decay time-constant to assign each point in delay embedded space (**B**) a phase $\theta$ (color of the manifold). $\theta$ identifies position of the system along the cyclic flux. Neuronal activity within a phase bin $\Delta\theta$ is averaged such that the center of mass of all points within a bin $\Delta\theta$ yields the coordinate of this bin in the delay embedded

space. This is used to construct a one-to-one map between a cloud of points in the delay embedded neuronal activity and a phase bin in the manifold space (number of points in each $\Delta\theta$ is shown by thickness). Thus, as the system evolves along $\theta$ in the direction given by $d\theta/dt$ (black arrows) it traces a trajectory in the delay embedded neuronal activity. Flux separation, accomplished using cluster analysis, assures that similar phases of different fluxes are not mistakenly grouped together (Materials and methods). (**F**) Simulations of the manifold in the phase space $(\theta, \alpha)$ recapitulate neuronal activation. To show this we compare the simulated distribution of time intervals between consecutive minima of A (return times) and those produced by the full system including both neurons (minima shown by black Xs in **A**).

DOI: https://doi.org/10.7554/eLife.46814.019

The following figure supplement is available for figure app11:

**Appendix 1—Figure 1 supplement 1.** State space and flux in dynamical systems with noise.

DOI: https://doi.org/10.7554/eLife.46814.020

Ultimately, we are interested in reconstructing the laws of motion that drive temporal evolution of neuronal activity. Although identification of the phase space is essential, it alone is not sufficient. Even in *C. elegans* the phase space is too high dimensional to estimate equations of motion directly from experimental data. To simplify the dynamics, we use the fact that in the limit of low noise, trajectories of the system will be dominated by motion near the loops. Therefore, although the phase space itself may be high-dimensional, most trajectories traced by the system in this space are low dimensional objects well-approximated by rotation around the loops referred to as cyclic fluxes. We introduce a novel method – Asymmetric Diffusion Map Modeling (See Materials and methods) – to extract these fluxes and create a parsimonious description of the dynamics. Diffusion maps express distances between points $i$ and $j$ in a local neighborhood of the phase space as the probability that a system found in state $i$ at time $t$ will be observed at state $j$ at time $t + 1$ (**Nadler et al., 2006**; **Coifman and Lafon, 2006**; **Lian et al., 2015**). This local neighborhood is defined by a Gaussian kernel centered at $i$. Accordingly, diffusion maps are symmetric and cannot capture the temporal sequence in which the data are observed. To adapt diffusion mapping for the purposes of modeling system dynamics, we center Gaussian kernels on the next experimentally observed point $\mathbf{X_{t+1}}$ instead of $\mathbf{X_t}$ (Materials and methods) (**Appendix 1—figure 1C**, red cloud shows the Gaussian kernel). Thus, although the system is most likely to transition to its next experimentally observed state, the Gaussian kernel ensures that the system may also diffuse within the neighborhood of $\mathbf{X_{t+1}}$. As a result the diffusion map is asymmetric and can express the ordered sequence of experimental observations. Diagonal bands in this transition probability matrix occur when the trajectory has parallel segments separated in time (**Eckmann et al., 1987**) (**Appendix 1—figure 1D**). This confirms the intuition that in the limit of low noise, the trajectories of the system are recurrent.

Expressing distances between points in phase space as transition probabilities offers a fundamental advantage – the dynamics given by the transition probability matrix can be simplified using spectral analysis. Spectral analysis decomposes the dynamics given by the original transition probability matrix into a sum of simple decay equations known as the eigenmodes **Appendix 1—figure 1D**. Because of the asymmetry in the diffusion matrix (**Appendix 1—figure 1D**), these eigenmodes given by the eigenvalue $\lambda_i$ and the associated eigenvector $\phi_i$ can be complex and thus reflect rotational motion. If the mode decays quickly, it does not contribute to the observed dynamics appreciably. Thus, dynamics of the system as a whole can be approximated by the complex eigenmodes with the slowest decay time-constant (Materials and methods). These slow decaying eigenmodes are an approximation of the cyclic motion around the loops in the phase space.

The position of each state of the system $\mathbf{X_t}$ along the cyclic flux is given by the phase $\theta$ of the associated complex eigenvector with the slowest decay time-constant (**Appendix 1—figure 1E**). In addition to $\theta$, the only other relevant variable is the identity of the flux, $\alpha$. Identify of the flux is established by clustering the diffusion map (Materials and methods). Because only two variables are sufficient to span the phase space, we can efficiently estimate equations of motion $\mathcal{M}(\theta, \alpha)$ from short and noisy experimental measurements.

$\mathcal{M}(\theta, \alpha)$ approximates the dynamics of the nervous system $f(\mathbf{X})$ in the true phase space $\mathbf{X}$ ($A$ and $B$ in this case). Yet, unlike the high dimensional and nonlinear dynamics given by $f$, $\mathcal{M}$ is two dimensional and linear. The nonlinearity of $f$ is encoded in the mapping from $(\theta, \alpha)$ to delayed coordinates $\langle A(t), A(t - \tau)...\rangle$ (Materials and methods). This mapping is illustrated in **Appendix 1—figure 1B and E**. Mapping from $(\theta, \alpha)$ back to neuronal activity can be used to validate dynamics given by $\mathcal{M}$. In **Appendix 1—figure 1F**, we used $\mathcal{M}$ of the two neuron system to predict the distribution of times between local minima of A (marked by the back Xs in **Appendix 1—figure 1A**). These correspond to the returns of the system to the same $\theta$ along one of the two fluxes.

The methodology illustrated for the two neuron system was applied to reconstruct the dynamics in *C. elegans*. Parameters used in the *C. elegans* model construction are shown in **Appendix 1—table 1**. We chose the delay embedding time τ based on the autocorrelations of neuronal activity and its derivative **Figure 2—figure supplement 2A**. Equivalently, this choice can be made on the basis of mutual information between delayed values of neuronal activity **Figure 2—figure supplement 2B**. The specific choice of τ and number of embeddings does not strongly affect the results so long as the total delay time is ~50 frames and the number of embeddings is sufficiently large **Figure 2—figure supplement 3**.

**Appendix 1—table 1.** Parameters used in model construction of *C. elegans* dynamics.

| Parameter name | Parameter value | Parameter description |
| --- | --- | --- |
| Number of neighbors | 12 | number of nearest trajectories in a local neighborhood |
| Minimum time frame | 50 | minimum time separation for nearest trajectories |
| Number of delays | 5 | Number of times the neuronal activity was embedded |
| Delay length | 10 | τ in units of number of frames |

DOI: https://doi.org/10.7554/eLife.46814.021

