## [Decision Letter]

Thank you for submitting your article "A model of conserved global neuronal dynamics predicts future behaviors in *Caenorhabditis elegans*" for consideration by *eLife*. Your article has been reviewed by Ronald Calabrese as the Senior Editor, a Reviewing Editor, and three reviewers. The following individuals involved in review of your submission have agreed to reveal their identity: William S Ryu (Reviewer #1); Elizabeth Cropper (Reviewer #2).

The reviewers have discussed the reviews with one another and the Reviewing Editor has drafted this decision to help you prepare a revised submission.

Summary:

This paper reports a "coarse" grain model that takes whole "brain," single-neuron calcium data from *C. elegans* and provides behaviorally relevant results using a manifold detection method based on diffusion mapping. The paper is technical enough to be of interest to specialists, but written in a way that is accessible and potentially interesting to a general audience. The Materials and methods sections is particularly clearly presented.

Essential revisions:

While there was considerable enthusiasm for the approach, there were several concerns that must be addressed before publication. The expert reviews are appended and will be of critical importance in the revision. The most important concerns are:

1) The data is from constrained animals and this impacts the interpretation of the results. Reviewers #1 (comment 1) and #3 (comment 1) share this concern and have specific prescriptions.

2) Not enough detail is provided about the model itself. Reviewers #1 (comment 2) and #3 (comment 2) share this concern and have specific prescriptions.

3) The time scale of the delay embedding is a concern and should be addressed as called for in comment 5 of reviewer #3.

4) There was a concern that predicting behavior based on the data that was used to define behavior is circular. To justify the conclusions the authors should show that the conclusions continue to hold when using the data without AVA (comment 3, reviewer #3).

5) There was a concern about neuronal identification in the data. This concern is difficult to address since the authors rely on published data and are not themselves doing the neuronal identification. The authors should combine a discussion of the robustness of their analysis with respect to neuronal mis-identification (especially considering that neuron identification for large scale recordings is still a major technical challenge for many labs) with a general robustness analysis for the previous point, where they would remove AVA and systematically explore how removal/identity shuffles would affect the resulting manifold and prediction. If this discussion and analysis is provided, the authors need not address reviewer #3, comment 4 further.

Title

The authors should consider revising the Title to reflect reviewer concern about whether the model is predictive.

*Reviewer #1:*

This is a very nice paper describing a "coarse" grain model that takes whole "brain," single-neuron data and provides behaviorally relevant results. The paper is technical enough to be of interest to specialists, but written in a way that is accessible and potentially interesting to a general audience. The Materials and methods section is particularly clearly presented. I think the work clears the bar for *eLife*.

1) Since the data are from constrained worms, it is not clear to the reader how these behavioral states were measured. A naive reader would assume behavior labeled as "forward locomotion," "reverse," etc. would come from observations of moving worms independently and not from interpreting the global brain signals themselves. Anyway, this can be made clear up front without asking the reader to go through Kato et al. For example, for Figure 2A, the authors explicitly write that they used the AVA signals to define the start of forward locomotion. What about the rest of the defined behaviors?

2) Not enough detail is given about the model in order for the reader to appreciate the jump from Figure 1 to Figure 2. The authors reference Figure 1—figure supplement 1 early in the Results sections but I would think something like Figure 4 would be necessary for the reader to be able to assess Figure 2. There should be enough technical detail given in the text so that the model is understandable.

3) Maybe this is a minor point (or an argument of semantics), but does the model really predict behavior of *C. elegans* up to 30 seconds in the future? Or does it predict the probability of a stochastic transition at some time T and so the event has some expected time, *t,* and manifests itself observably at *t* on average. The Title of the paper reads as if the signals deterministically predict future behavior.

4) A natural question is raised when discussing the number of neurons needed to see similar global brain dynamics. From 100 neurons to 15 neurons to 8 neurons. How far can one go for this specific dataset?

Discussion section. "*C. elegans* do not fire action potentials." Not strictly true. For example, see: Liu et el., 2018

*Reviewer #2:*

This report takes advantage of the powerful tools that have been developed that make it possible to relate neural activity to behavior in *C. elegans*. Namely, imaging techniques with single neuron precision can record activity in intact worms as they freely switch between different forms of locomotion. It is therefore possible to do more than simply correlate an activity pattern with a behavior. The temporal evolution of behavior can be characterized. There are not many systems where this can be accomplished, and this is a very exciting area of research. A potential 'drawback' of experiments like this that generate so much data is that data can be difficult to analyze and interpret. Studies such as this that develop tools for this purpose are therefore clearly needed.

These authors use imaging data to construct a model of neuronal dynamics. Their approach is novel, and differs from traditional approaches in that it does not proceed in a 'bottom-up' fashion (it was not built by characterizing all of the biophysical properties and synaptic connections of the neurons in the network). There are a number of drawbacks to the bottom-up approach, as the authors point out. For example, an assumption usually inherent in this type of work is that a particular network output is encoded by one set of circuit parameters. Work in other systems has indicated that this is not necessarily the case, and the authors demonstrate that activity in identified neurons in *C. elegans* is variable during the behaviors studied.

The tools that the authors develop extract information from a subset of the neurons that mediate behavior. There are hundreds of neurons in *C. elegans* but the authors were only able to consistently identify fifteen. This speaks to the potential utility of this method since it is generally not possible to record from all of the neurons in a network of interest. This is, however, not simply a 'methods' paper. The authors use their techniques to simulate neuronal activity and interestingly demonstrate that these simulations can be used to predict behavioral switches before they occur in a different cohort of animals (i.e., not the animals used to develop the model). Finally, the authors construct manifolds using specific data sets (e.g., activity of the fifteen identified neurons recorded from in four out of the five animals of the study) and demonstrate that left out data are well approximated by these manifolds. Taken together, this research comes to a fundamentally important conclusion – that global dynamics in the functioning of the nervous system are conserved despite the fact that there are differences in the activity of individual neurons.

*Reviewer #3:*

The authors of "A model of conserved global neuronal dynamics predicts future behaviors in *Caenorhabditis elegans*" re-analyze existing wholebrain calcium data from *C. elegans* using a manifold detection method based on diffusion mapping. Based upon the current manuscript I have a few concerns that if addressed would significantly improve the clarity of the manuscript.

1) My main concern is the interpretation of the results as predicting behavior: From the Title and the main text it is unclear to the reader that the animals in question are actually immobilized (according to Kato et al. these animals are in microfluidics and sometimes even treated with a paralytic). Kato et al. show that animals where AVA is silenced do not perform any reversals, but the global brain dynamics are still observed. This indicates a loose connection between these manifold dynamics and behavior at best. Kato et al. also reported that prolonged activation phases of neurons such as RIM only occur in immobilized animals, not in freely moving ones, indicating that immobilization changes neural dynamics. Based upon these caveats in their data, I urge the authors to carefully re-word their interpretation of their results as 'behavioral coding', and be more careful about this wording throughout, but particularly in the Discussion and the Title.

2) Reading the paper, it is unclear if the authors main goal is to present a method or to describe new findings. If the main goal is to present a generalizable method, the authors should be much more explicit about the steps of the data analysis process. In the current manuscript, the model is not described in the main text at all (subsection “ng neuronal dynamics give rise to neuronal activity” introduces the model without describing any of its properties). In either case, I strongly urge the authors to either present a cartoon or an example data set that underwent all of their processing, embedding and dimensionality reduction, etc. If I read this manuscript as a methods paper, I would like to see how parameter choices (in particular delays, smoothing parameters, numbers of dimensions chosen after reduction) affect the outcome. This could be done on purely synthetic data even.

3) As far as I can tell from the methods, the 'behavior' is deducted from the activity of the motor command interneuron AVA. AVAL and AVAR also appear among the 15 neurons that are common between datasets. It seems that AVAR and AVAL were not removed from the data used to create the manifold. Predicting behavior based on the data that was used to define behavior seems circular. It would strengthen the conclusions if they were still true from the data without AVA.

4) Neuronal identity: The analysis by Brennan and Proekt relies on unambiguous identification of the neuronal identity. The conclusions about variability in activity between neurons (Discussion section), and the fact that PCA does not create reliable manifolds could possibly indicate that a subset of the neurons were mis-identified. On a subset of only 15 common neurons, even one misidentified neuron could possibly have a large impact. From the periodic, low dimensional example dataset shown in Figure 1, and the somewhat consistent PCA weights shown in Figure S2 of Kato et al. this conclusion is surprising and could be better supported to motivate the more complex strategy presented in the paper.

Relatedly, one of the first findings presented in Figure 1 is that there are consistent statistical differences in the activity of the same neuron across animals. Based upon single neuron Calcium imaging, it is not surprising that neurons are showing diverse activity in 'behaviors' they do not control. Comparing these data with previous studies on variability in neural activation (Gordus et al., 2015 for example) could provide context for these observations.

5) Timescales: Delay embedding is highly sensitive to the chosen timescales of the delay(s). The authors used a delay of ~4 seconds. However, the highly periodic nature of the neural activities (see e.g. Figure 1A, cyclic activity in most neurons) means that the auto-correlations will also have periodicity and signals will have non-zero auto-correlation over significantly longer times. The authors could show the auto-correlations explicitly and show how the delay embedding changes with significantly longer delay times or using a different method such as mutual information to calculate the delay. I suspect the signals have auto-correlation times much longer than 30 seconds.

---

## [Author Response]

Essential revisions:While there was considerable enthusiasm for the approach, there were several concerns that must be addressed before publication. The expert reviews are appended and will be of critical importance in the revision. The most important concerns are:1) The data is from constrained animals and this impacts the interpretation of the results. Reviewers #1 (comment 1) and #3 (comment 1) share this concern and have specific prescriptions.

This is a very important point. We have included more details concerning specifically the fact that the models of neuronal dynamics were constructed on the basis of neuronal activity observed in immobilized worms. Thus, the best interpretation of observed neuronal activity is that it expresses sequences of motor commands that occur in the absence of actual execution of behaviors. We have changed the Abstract, Title, and the Discussion section to reflect this. For the construction of the model we adapted the behavioral state definitions used by Kato et al. We focused our predictions on backwards and forward locomotion because they are by far the most prevalent in the observed time series. In principle, our methodology can be used on the other behaviors as well, but these predictions will be noisier due to paucity of observations.

There are likely to be important differences between neuronal dynamics observed in the immobilized and freely moving animals. We now dedicate a section in the discussion specifically to this issue. Differences in neuronal dynamics in freely moving and immobilized worms can be of two fundamentally distinct types: quantitative and qualitative. It is true that freely moving worms do not exhibit as much backward locomotion as those immobilized in the microfluidic chamber. Also, backward locomotion episodes are briefer in freely moving animals. These constitute quantitative differences. All that would be required to adapt our manifold constructed on the basis of immobilized worms to freely moving ones is to change the probability of switching between different loops in the manifold and the phase velocity of progression along the loop. In this case, the overall shape of the manifold would not change. It is also possible that execution of motor behaviors and sensory feedback may change the shape of the manifold appreciably. This would constitute a qualitative change. Neuronal suppression experiments by Kato et al. suggest that this may not be the case in *C. elegans* locomotion as they observe that even when backward locomotion was eliminated by suppressing activity of the AVA, other neurons involved in backward locomotion were activated in a similar fashion. Yet, to definitively determine whether the shape of the manifold is significantly altered by sensory experience and execution of behaviors, one would need to reconstruct neuronal dynamics in freely moving animals. Our methodology will be useful for accomplishing this important goal in the future.

In simple systems it is known that many behaviors are controlled by central pattern generators. Motor programs observed in isolated nervous systems are essentially similar in terms of phasing of activation of individual neurons to those observed during natural execution of behaviors. Interactions with the environment may change the speed or the finer details of the motor programs observed in the isolated brain. Our model of neuronal dynamics in immobilized worms can be thought of in similar terms. The dynamics express an action sequence of different locomotor behaviors observed in the nervous system isolated from the environment. This action system is structured by neuronal dynamics which we reconstruct on the basis of neuronal activity.

2) Not enough detail is provided about the model itself. Reviewers #1 (comment 2) and #3 (comment 2) share this concern and have specific prescriptions.

We have included a new section and included it into the Appendix. This newly added section and figures describe the basic ideas behind the methodology. To illustrate how the method works, following the suggestion of reviewer 3, we simulated a network of two interconnected neurons and reconstructed the dynamics using our methodology based on one of them (Appendix 1—figure 1 and its corresponding Appendix 1—figure 1—figure supplement 1). Much like the nervous system of *C. elegans* the dynamics of this simple system are nonlinear and noisy. The reconstruction of the model system includes all of the essential elements of the modeling approach including delay embedding, construction of the asymmetrical diffusion map, spectral analysis, and clustering. We then illustrate how predictions of the model can be verified by comparing the statistics of the reconstructed system to those observed in simulations of the complete neuronal network. We have also included a table that summarizes all the parameters used to model *C. elegans* dynamics (see also our response concerning the choice of embedding parameters).

3) The time scale of the delay embedding is a concern and should be addressed as called for in comment 5 of reviewer #3.

The delay embedding time was chosen empirically based on the scale of autocorrelation of activation of individual neurons now shown in a newly added supplementary figure (Figure 2—figure supplement 2). The choice of the embedding parameters (delay time and number of embeddings) can, as expected, affect the quality of the reconstruction. To address this concern, we varied embedding parameters and included the results of this analysis in a newly generated figure (Figure 2—figure supplement 3). To determine the quality of the reconstruction we compared the dwell time statistics of models constructed using different values of delay times and number of embeddings to those observed experimentally by Kato et al. These results show that several parameter choices yield good predictions and established some parameter choices that do not. We now discuss this in the Materials and methods section.

Other measures such as mutual information can also be used to determine delay embedding parameters. We also included delay mutual information in a supplementary figure as suggested by the reviewer. Because of paucity of data, we used autocorrelations in this manuscript as the basis for choosing delay embedding parameters. In other systems mutual information may prove to be more useful. In general, the issue of choosing appropriate embedding parameters in multivariate and noisy time series is not well resolved. Nevertheless, because our method is able to reduce the dimensionality of the embedded time series, several choices of the number of embeddings yield good predictions so long as the number of dimensions is sufficiently high, and the delayed values of the data set are approximately independent.

4) There was a concern that predicting behavior based on the data that was used to define behavior is circular. To justify the conclusions the authors should show that the conclusions continue to hold when using the data without AVA (comment 3, reviewer #3).

We have eliminated AVA from both the Kato et al. dataset used for model generation and from the Nichols et al. validation dataset. In a newly added supplementary figure (Figure 3—figure supplement 1), we show that even in the absence of the AVA our model can predict the expected switching time between motor commands for forward and backward locomotion. These predictions are nearly as accurate as those obtained on the full neuronal subset. Predictions given by the model in the absence of the AVA are significantly more accurate than those given by the model that relies solely on dwell time statistics.

As pointed out by reviewer 1, the diffusion map is a stochastic model as it is based upon a Markov matrix. Thus, the predictions are best described in terms of the expected time to motor command switch. We now make this point explicitly in the manuscript. The important finding is that this expected time to switch of the motor command is a function of phase along the manifold rather than specifics of activation of individual neurons.

5) There was a concern about neuronal identification in the data. This concern is difficult to address since the authors rely on published data and are not themselves doing the neuronal identification. The authors should combine a discussion of the robustness of their analysis with respect to neuronal mis-identification (especially considering that neuron identification for large scale recordings is still a major technical challenge for many labs) with a general robustness analysis for the previous point, where they would remove AVA and systematically explore how removal/identity shuffles would affect the resulting manifold and prediction. If this discussion and analysis is provided, the authors need not address reviewer #3, comment 4 further.

Our model is based upon previously acquired data and we cannot offer any additional information concerning neuronal identification beyond that presented by Kato et al. and Nichols et al. It is possible that some neurons were misidentified. Thus, we only focused on the 15 neurons that were identified with highest confidence in the Kato et al. dataset. The data in Figure 1 argue that neuronal misidentification is not likely a strong contributor to the variability of neuronal activation. For instance, activation of RIML (highlighted for the purposes of illustration in Figure 1) is inconsistent in some forms of locomotion but is consistent among worms in another type of locomotion. This is the case in most neurons identified by Kato et al. To further address the issue of neuron identification and also a question raised by reviewer 1 concerning the minimal number of neurons required for the reconstruction, we built models of dynamics on the basis of activity of a single neuron. The results of this analysis is included in a newly created supplementary figure (Figure 2—figure supplement 4). These results indicate that some single neurons yield predictions comparable to those observed for the entire 15 neuron set. Remarkably one of the neurons that is sufficient for good quality prediction (~ 75% as informative as the entire set of 15 neurons) is RIML. Recall that RIML exhibited variable activation during backward locomotion (Figure 1). Nevertheless, predictions based solely on RIML are some of the more informative among models based on single neurons.

Regarding the variability of neuronal activation in behaviors that they do not control, we specifically discuss the ALA neuron in the manuscript. Note that activation of ALA is variable from one cycle of behavior to the other. This is reflected in broad confidence intervals around the mean activity of the ALA. Note, however, that there are no consistent differences between ALA activity observed in different animals. The consistent differences between individuals are of fundamentally different nature than cycle to cycle variability. In each individual, the neuronal activity is reproducible from cycle to cycle. Yet, there are consistent differences between worms. These differences are such that averaging neuronal activity across worms does not yield a representative trace of neuronal activity. This kind of variability is the focus of our manuscript. Gordus et al. focus on how taking into account variable activation of other neurons in the circuit constraints activation of the AIB neuron. This is in spirit similar to our argument that while there are individual differences in neuronal activation, the macroscopic parameter such as phase along the cyclic flux nevertheless expresses dynamics that are consistent among individuals. Thank you for pointing out this study to us. We now cite it explicitly in the manuscript.

TitleThe authors should consider revising the Title to reflect reviewer concern about whether the model is predictive.

The new Title is: A quantitative model of conserved macroscopic dynamics predicts future motor commands.